# Uncertainties in measuring offshore gambling: A scoping review of Nordic approaches

Virve Marionneau[1]*, Søren Kristiansen[2], Tomi Roukka[3], Håkan Wall[4,5]

**1** Centre for Research on Addiction, Control and Governance, University of Helsinki, Helsinki, Finland, **2** Department of Sociology and Social Work, Aalborg University, Aalborg, Denmark, **3** Finnish Institute for Health and Welfare, Helsinki, Finland, **4** Department of Clinical Neuroscience, Centre for Psychiatry Research, Karolinska Institute, Stockholm, Sweden, **5** Stockholm Health Care Services, Region Stockholm, Stockholm, Sweden

* virve.marionneau@helsinki.fi

## Abstract

### Background

A part of online gambling consumption takes place in offshore markets. Lack of regulatory control over offshore offers erodes many public policy and public health objectives. Channelling consumption from offshore markets to regulated markets is therefore used as justification in many gambling policy decisions. Yet, it is currently unknown how reliable existing estimates of the size of offshore markets are.

### Methods

This scoping review investigates how offshore gambling markets are measured in the Nordic context and what kinds of uncertainties are involved in existing measures. We searched available estimates of offshore gambling markets from academic and grey literature in four Nordic countries (Denmark, Finland, Norway, Sweden). The final sample consists of 32 reports. To supplement the results, we conducted key informant interviews and our own analysis of available data.

### Results

24 estimates concerned the monetary value of offshore gambling as a percentage of the full market. Nine estimates concerned the population prevalence of offshore gambling. In terms of methodologies, most studies reported figures from a private gambling intelligence company H2 gambling capital, either directly or combined with other data sources. Different methodological choices yielded different estimates. An important part of reports was funded by the gambling industry. Industry reports tended to have higher overall estimates of offshore gambling due to methodological choices.

**Data availability statement:** All data used in the scoping review are fully available from public sources and uploaded in OSF repository (https://osf.io/67n3h/). Data from H2 gambling capital are only available under license. These data can be made available by the authors with permission from H2 (https://h2gc.com/ ; email: data@h2gc.com). Key informant interviews are not publicly available since we did not obtain a permission from these individuals to quote them directly in the manuscript nor to share the transcription publicly. The data are described in a University of Helsinki data repository including information on how to access these data (https://doi.org/10.60668/hulib:12).

**Funding:** VM and TR received funding from the Finnish Ministry of Social Affairs and Health based on provisions of the Finnish Lotteries Act §52. SK is funded by the University of Aalborg. HW received funding from the Swedish Research Council for Health, Working Life, and Welfare (Forte, grant number 2023-00898). Open access funded by Helsinki University Library. The funders had no role in study design, data collection and analysis, decision to publish, or preparation of the manuscript.

**Competing interests:** The authors have declared that no competing interests exist.

## Conclusions

The measurement of offshore gambling is a politically sensitive topic wrought with uncertainties. More reliable methods and figures are needed to better inform harm prevention and consumer protection in the online environment. Inaccurate offshore measures can be used as a tool for regulatory resistance. A transparent and scientifically validated measurement tool is needed to improve the evidence-base.

## Introduction

Offshore gambling refers to various forms of online gambling provision that emanate from outside the point of consumption jurisdiction. Offshore gambling is provided without an operating license within that jurisdiction [1]. Any unlicensed cross-border offer can, in practice, constitute offshore gambling. Offshore markets are considered 'grey', when offers are not legally prohibited nor legally permitted in a jurisdiction. Offshore markets are 'black' when provision is illegal [1]. Within the European context, grey market operations include gambling offers that are licensed in another country within the European Economic Area. Black market refers to any operator providing unlicensed gambling in Europe. Offshore gambling operators are able to challenge and circumvent regulations in point of consumption jurisdictions [2]. This weakens the reach and effectiveness of national regulatory responses.

Offshore gambling can pose a challenge to effective regulation: Offshore jurisdictions generally provide weaker regulations regarding consumer protection, data privacy and age limitations. Offshore jurisdictions may also be less stringent in terms of preventing criminal involvement in gambling provision, or in implementing anti-money laundering interventions. [2–7]. Offshore offers also increase competition within the market – this can increase the availability and visibility of gambling in societies, leading to increased levels of harm. Point-of-consumption regulators have no legal means to control harmful product characteristics, age restrictions, or targeted advertising conducted by offshore operators. Furthermore, offshore websites are available for individuals who self-exclude from their national gambling offer. Effective regulation of offshore gambling is therefore a question of public interest and of public health. Previous research on offshore gambling has found that gambling on offshore websites is connected to higher levels of problematic gambling and gambling-related harms than gambling on regulated offers [8,9].

### Channelling policies and measures

Offshore gambling is closely linked to the idea of channelling. Borch [5:235] has defined channelling as a tool employed by governments 'to direct gamblers from unlicensed (mainly offshore) to licensed (mainly national) games' for national interest. Channelling has become a key policy objective in many jurisdictions [2,5,10]. Channelling has also been used as a rationale for policy changes. Several countries have either introduced regulated online gambling markets or shifted to license-based models to increase their control over gambling markets and to channel consumption to regulated offers [11].

A range of policy tools are available to channel consumption to the regulated market. Many of these are restrictive, aiming at deterring offshore provision or demand. These include blocking payments and web traffic to offshore websites, enforcement action against unregulated providers, a range of information tools to raise awareness, collaboration with online platforms to reduce visibility, and B2B licensing to limit the software options available to offshore companies [2,12]. Another approach to channelling has consisted of making the regulated market more attractive. This can be accomplished, for example, by allowing extensive marketing and product development of legal offers [4]. The effectiveness of any of these measures is difficult to assess [2]. This has been because, despite the political importance of channelling, it is surprisingly unclear how and if we can measure developments in the unregulated market.

There is an emerging body of international research attempting to measure the size of offshore gambling markets. Most of this research is based on survey studies conducted amongst the general population or help-seekers, focusing on the population prevalence of offshore gambling participation [8,9,13,14]. Results from these studies have differed significantly depending on the legality of online gambling in the national market and across product groups.

The monetary value of offshore gambling is also difficult to measure, although some market intelligence companies such as H2 Gambling Capital do produce estimates. There are currently over 6,000 active gambling websites [15], many of which do not disclose their operational metrics, income statements or the jurisdictions in which they operate. Offshore gambling companies also differ significantly from each other. Some black-market operations are connected to organised criminality [16] while most grey market actors include companies licensed in 'offshore jurisdictions', such as Malta and Gibraltar [1,3,17]. The availability of operational data from these different types of companies varies, as many are not mandated to disclose any business data.

## The current study

Inherent difficulties and epistemic uncertainties involved in measuring the offshore market are reflected in a variety of methodological approaches and data sources that can result in widely varying estimates. The current paper produces a scoping review on how offshore gambling has been measured across four Nordic countries (Denmark, Finland, Norway, and Sweden) and what kind of uncertainties are involved in measuring the size of the offshore market segment. The focus on the Nordic markets is justified by the high level of online gambling participation in the Nordics and well-established gambling market and gambling harm monitoring systems. Nordic countries lead the statistics in terms of highest online gambling participation in Europe and were among the first to provide legal online gambling. The Nordic countries also share many other societal and economic similarities. As offshore measurements are highly sensitive to contextual issues, it is more suitable to compare relatively similar societies. All Nordic countries have gambling regulators that are tasked with measuring and regulating the offshore market, including provisions to block offshore offers, via payment blocking in Finland and Norway, wallet-to-wallet-based payment blocking in Sweden, and website blocking in Denmark. Norway also actively enforces against offshore operators by issuing fines. While details of gambling regulation vary across the Nordic countries, each country also puts significant emphasis on 'channelling' in national gambling legislation [18–21].

In this review, we focus on how offshore market sizes have been measured in the Nordics and what kind of methodological uncertainties are involved in this field. We produce a scoping review of available evidence on channelling rates and methodologies used to produce these in the Nordics. We focus on the offshore share of spending as a percentage, share of spending in monetary terms, and share of population prevalence. Our aim is to identify how channelling rates have been measured, by whom, and what kind of estimates have been obtained. Measurement of offshore gambling market sizes has concrete policy implications, as these results are widely used to justify gambling policy choices in the Nordic region and beyond. We also identify existing gaps and produce recommendations for more systematic monitoring of offshore gambling to improve prevention of harmful online practices.

## Materials and methods

In line with the scoping review methodology [22,23], we reviewed academic studies, government reports, and other documentation presenting an estimation of offshore gambling in Denmark, Finland, Norway, or Sweden.

A scoping review approach is justified by lack of standardisation within the field of offshore measurement, and our interest in uncertainties involved in measuring offshore gambling. Scoping reviews allow examining the range and nature of evidence and summarising findings from a heterogenous body of evidence [22]. Existing studies on offshore gambling use various, sometimes incompatible methodologies. It is therefore crucial to critically map available methods and approaches rather than to produce a systematic meta-analysis of existing results that are likely to be highly contested, lack accuracy and therefore impossible to validate. Unlike systematic reviews, scoping reviews aim at presenting 'a broad overview of the evidence pertaining to a topic, irrespective of study quality, and are useful when examining areas that are emerging, to clarify key concepts and identify gaps' [22: 2]. In line with this definition, our aim was to map what kind of methodologies are used to measure offshore gambling or channelling, compare available estimates and to identify uncertainties involved in existing approaches to monitor the field.

We complement the scoping review with other methodologies, including an analysis of key informant interviews and an analysis of data derived directly from gambling data intelligence provider H2 gambling capital. This allows us to look at the measurement of channelling using three key metrics: share of spending as a percentage, share of spending in monetary terms, and share of population prevalence.

Our review focused on the Nordic context, only, for two reasons. First, the Nordic contexts provide a natural comparative setting of countries with differing gambling systems (licensing, monopoly) but similar societal contexts. Second, as the measurement of offshore gambling is highly political and contested, it was important to study contexts that were familiar to the research group. This allowed us to identify and understand potential biases, even though the quality of included studies did not permit systematic evaluation of risk of bias across included reports. We were not able to validate or cross-validate the accuracy of data sources because there is no current gold standard within the field, complicating the choice of comparative points. Instead, the aim was to use the scoping methodology to investigate uncertainties in this field.

## Study selection and retrieval

First, we identified relevant studies by conducting a literature search in scientific databases. As academic literature in the Nordic area is widely published both in English and national languages, we first conducted the search in English on Scopus, Ebscohost, and Google Scholar. We conducted the same search in Nordic languages (Danish, Finnish, Norwegian, Swedish) on Google Scholar. The keywords used to conduct the English language searches were: 'gambling + offshore + measur*/market + Finland/Sweden/Denmark/Norway'. The equivalent searches in Danish were: pengespil/gambling + offshore/kanalisering + andel + marked'; in Norwegian: 'pengespill + offshore + kanalisering + dele + marked'; in Finnish 'rahapel*+offshore/"järjestelmän ulkopuolinen" + mitta*/markkina'; and in Swedish '"spel om pengar"+offshore/kanalisering+andel/marknad'.

To capture grey literature, we also conducted the searches on Google as well as consulting national authorities and databases for any additional resources. The included national databases are described in Table 1. For any annual reports, we only included the most recent outputs. As for searches on Google Scholar and Google, we only included the first 20 hits in cases where more hits were found. We set the limit at 20 after a preliminary screening establishing that further results provided no relevant sources. This limit was validated by two members of the research team.

As our focus was on uncertainties within and across countries and methodologies, we limited the search to all types of documents published between January 2010 and March 2024 when the searches were conducted.

The identification, screening, and inclusion of studies is described in Fig 1, adapted from the PRISMA 2020 guidelines [22]. The PRISMA 2020 checklist is provided in the supplementary material. Our search yielded a total of 679 records. Based on the titles of references, we first removed the duplicates (N = 253) and unrelated hits (N = 302). Unrelated hits

**Table 1. National databases and websites included in the scoping review.**

| Country | Websites and databases | Description |
|---------|------------------------|-------------|
| Denmark | Danske Spil | Former monopoly operator |
|         | Spillebranchen | Industry organisation |
|         | Spillemyndigheden | Gambling regulator |
|         | Spiludbydere | Affiliate marketing site |
|         | VIVE | Centre for social science research |
| Finland | KKV | Competition Authority |
|         | Poliisihallitus | Gambling regulator |
|         | Suomalainen rahapeliyhdistys RY | Information site |
|         | THL | Public health institute |
|         | Veikkaus | Monopoly operator |
| Norway | Actis | NGO umbrella organisation |
|        | Lotteri og Stiftelsestilsynet | Gambling regulator |
|        | Norsk Rikstoto | Monopoly operator |
|        | Norsk Tipping | Monopoly operator |
|        | Norske spilleautomater | Affiliate marketing site |
| Sweden | ATG | Former monopoly operator |
|        | BOS | Industry organisation |
|        | Fakta om Spel | Information site |
|        | Folkhälsomyndigheten | Public health institute |
|        | Spelinspektionen | Gambling regulator |
|        | Statskontoret | State treasury |
|        | Svenska Spel | Former monopoly operator |

included, for example, studies on offshore banking or offshore drilling, or gambling-related studies that were focused on individual-level gambling behaviour rather than markets.

## Study screening

We screened the remaining 124 records for their relevance based on their abstracts, resulting in the exclusion of a further 58 papers. Screening and all inclusion decisions (including preliminary screening) were made in agreement by VM and SK. We were unable to access two documents, bringing the number of full papers or reports read to 64. Both at the abstract screening and full report reading phase, we applied the following inclusion and exclusion criteria:

The inclusion criteria were that (1) The paper or report produces an original estimate of offshore gambling OR (2) the paper or report presents an estimate of offshore gambling as part of policy development. Furthermore, (3) the paper or report needed to focus on the Nordic countries.

Correspondingly, the exclusion criteria were that (1) the paper or report did not focus on the Nordic region; (2) prior reviews; (3) studies focusing on offshore gambling of subpopulations such as treatment-seekers or self-excluded instead of full population measures; (4) papers or reports where the methodology for obtaining an offshore estimate was not described; and (5) papers or reports that did not provide an offshore estimate. Often, studies only referenced an estimate in their discussion sections, but the focus of the paper was not to measure offshore gambling. The number of excluded papers at both stages and per criterion are described in the PRISMA flow chart (Fig 1). The final sample consists of 32 papers and reports (see Table 2 in the results section). All reports underlying the scoping review analysis are available publicly in the following repository: https://osf.io/67n3h/.

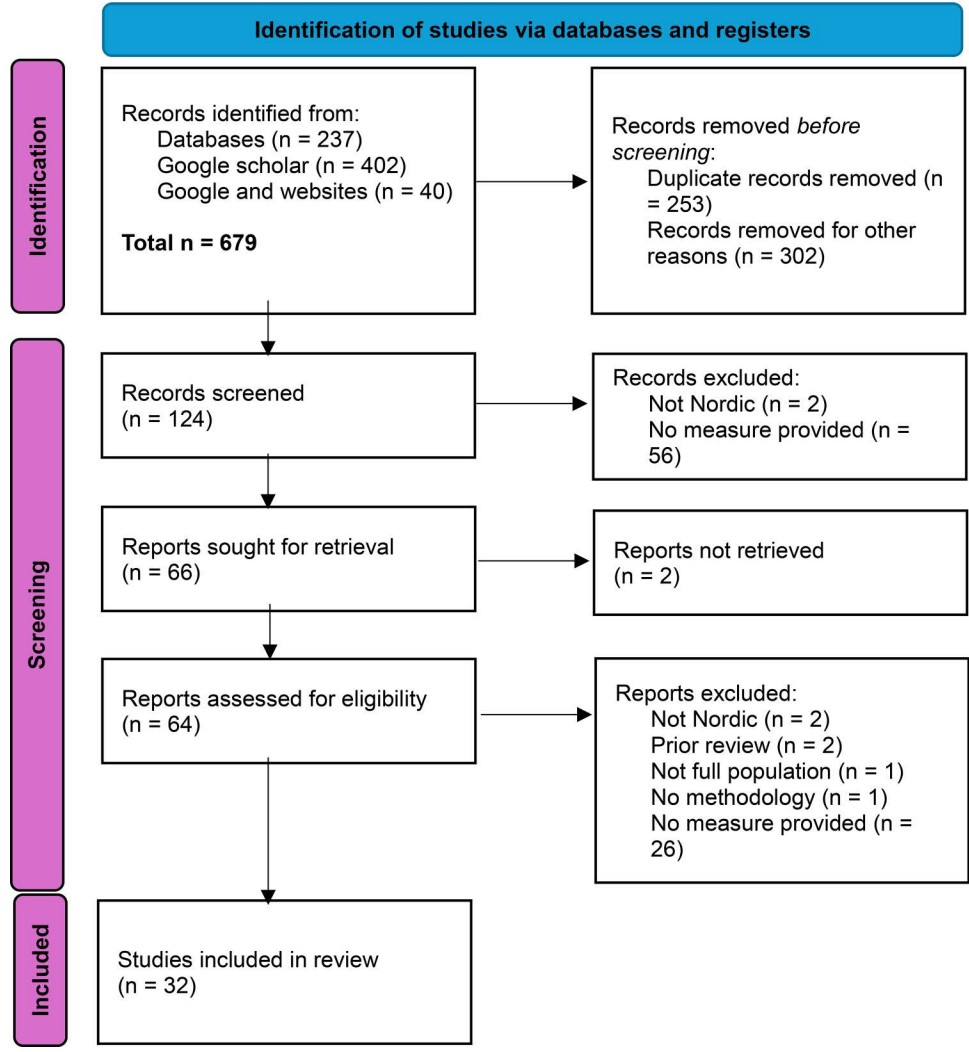

**Fig 1. Prisma flow diagram.**

## Additional data sources

### Key informant interviews

To inform the analysis, we also consulted local authorities in Denmark, Finland, Norway, and Sweden on their on-going approaches to measuring offshore gambling. We conducted a thematic key-informant group interview in an online meeting with authorities from Denmark, Finland, and Sweden, and via email with an authority from Norway. All participants represented national authorities and regulators who were either involved with the issue of offshore gambling measurement or using these data in their work. All participants were recruited via existing contacts based on their position and expertise on offshore gambling. The key informants included employees with the local gambling regulators with a responsibility for offshore gambling markets (Denmark, Sweden, Norway) and employee with the competition authority with a focus on offshore gambling markets (Finland).

**Table 2.  Studies and reports included in the review.**

| Reference | Data years | Context | Funder | Measured | Data source |
|---|---|---|---|---|---|
| [24] | (2010)–2013 | Norway | Government (Ministry of Culture) | Spending | H2 data |
| [25] | 2019-2023 | Sweden | Industry (ATG) | Spending | Web traffic analysis |
| [26] | 2016 | Sweden | Industry (BOS) | Spending | Survey of at least quarterly gamblers (N = 1,014) |
| [27] | 2020 | Sweden | Industry (BOS) | Spending | Sales figures, survey (N = 334) |
| [28] | 2020 | Sweden | Industry (BOS) | Spending | H2 data, tax data, survey (N = 1,000), indicators of total gambling |
| [19] | 2019 | Finland | Government | Spending and population share | Population survey (N = 3,077 past year gamblers), H2 data |
| [29] | 2011-2017 | Norway | Industry (Kindred, Gaming Innovation, ComeOn, Betsson) | Spending | H2 data, projections using spending data |
| [30] | 2012-2020 | Denmark | Industry (DOGA) | Spending | H2 data |
| [31] | 2010-2019 | Sweden | Industry (BOS) | Spending | H2 data, survey of providers |
| [32] | 2019 | Finland | Government (Ministry of Social Affairs and Health) | Population share | Population survey (N = 3,077 past year gamblers) |
| [33] | 2022 | Norway | Government (Lotteri- och stiftelsestilsynet) | Spending and population share | Sales data, quarterly survey (N = 1,000), H2 data, GBGC data, Swedish tax data |
| [34] | 2021-2023 | Norway | Government (Lotteri- och stiftelsestilsynet) | Spending and population share | Sales data, survey (N = 2,000), H2 data, GBBC data, Swedish tax data |
| [35] | (2010)-2014 | Norway | Industry (Betsson, Cherry Group, ComeOn, Guts, Unibet) | Spending | H2 data |
| [36] | 2020-2021 (spending); 2015–2021 (population share) | Norway | Industry (Norsk Rikstoto) | Spending and population share | Figures from regulator and industry |
| [37] | 2020 | Sweden | Government (Spelinspektionen) | Population share | Survey (N = 1,139 past year gamblers) |
| [38] | 2018 | Norway | Government (Ministry of Culture) | Spending and population share | Sales data, survey (N = 1,000), key informant interviews and documents |
| [18] | (2010)-2015 | Sweden | Government | Spending | H2 data, data from regulator |
| [39] | 2022 | Finland | Government | Spending | Finnish competition authority estimate |
| [40] | 2019 | Finland | Government (Ministry of social affairs and health) | Spending, population share | Population study (N = 3,122 past year gamblers) |
| [41] | 2021 | Sweden | Government (regulator) | Population share | Survey (N = 3,208 of whom 1,002 past 3-month web gamblers) |
| [42] | 2023 | Sweden | Industry (BOS) | Spending | Survey (N = 9,850 of whom 3,000 past 3-month gamblers). |
| [43] | (2010)-2013 | Sweden | Government (regulator) | Spending | H2 data |
| [44] | 2012-2016 | Sweden | Government (regulator) | Spending | H2 data, own analysis |
| [45] | 2017-2021 | Denmark | Government (regulator) | Spending | H2 data |
| [46] | 2022-2023 | Denmark | Government | Spending | H2 data |
| [47] | 2024 | Denmark | Government | Population share | Survey (N = 7,636) |
| [48] | 2019-2020 | Sweden | Government | Spending | H2 data, stakeholder interviews |
| [49] | 2019-2020 | Sweden | Government | Spending | H2 data |
| [50] | 2021–2024 | Finland | n/a | Spending | H2 data with 'specifications' |
| [51] | 2021 – 2024e | Finland | Government (Ministry of Social Affairs and Health) | Population share | Monthly survey (N = 2,000) |
| [52] | 2022 | Finland | Academic | Population share | Survey (N = 1,075) |
| [53] | 2023 | Finland | Industry (Veikkaus) | Spending | H2 data |

The consultations were centred around four themes: (1) reasons for reregulating gambling under a licensing regime or for maintaining a monopoly regime (including role of channelling); (2) How the offshore market is measured; (3) Why the offshore market is measured; (4) Cooperation around measurement and channelling in the Nordics. The interviews were conducted by TR and HW.

### Data from H2 gambling capital

As our analysis indicated that data from market intelligence company H2 gambling capital is widely used across the Nordic countries to estimate offshore gambling market shares, we also conducted our own analysis using these data. TR had access to data from H2 via license to the Finnish Institute for Health and Welfare. For the analysis, we tracked annual H2 estimates of grey market offshore gross gambling revenue (GGR) in Sweden, Finland, Denmark and Norway from 2022-10-01–2024-05-01. This time frame is shorter than for the scoping review due to lack of access to prior data. According to H2, black market shares (i.e., gambling companies outside of the EU) were not included in their estimates at the time of our data collection. Some of these actors can be connected to criminal activity and operate outside of any possibility of channelling towards legal systems. At each measurement occasion, H2 makes predictions on country-level offshore GGR based on most recent information.

## Analysis methods

### Scoping review

For each included study, we noted the reference, year, context (country), funder, what was measured (spending, population share), data sources used, methodological description for how offshore gambling was measured, and given estimates from 2010 onwards. All screening was conducted in collaboration by VM and SK to ensure that data were checked for accuracy. VM extracted the data first and SK double-checked them for accuracy. The included variables in the screening were decided by the research team based on an initial reading of included documents. The variables were chosen to reflect our original research aim which was to identify how and by whom estimates on offshore market sizes are produced, what kind of estimates have been produced and what kind of uncertainties are involved in measurement.

We did not produce a risk of bias estimation on the studies because this is not standard practice in scoping reviews. Furthermore, the included studies ranged in terms of methodologies, and many lacked methodological information. Some studies used a population study methodology, while others used data from H2 gambling capital or other market estimators. These varying methods would have required different tools for risk of bias estimation, reducing comparability. Furthermore, most studies in our review included little details on methodological choices, further reducing our possibility to objectively assess the quality of included studies or the robustness of their measures.

The analysis was conducted separately for GGR-based estimates and population share-based estimates. The reason for this was that these estimates measure different things and cannot be compared. Visualisation of results in plots were produced with R.

### Key informant data

The interview data were analysed using deductive qualitative content analysis, informed by our findings from the scoping review. Deductive content analysis is a well-suited approach for confirmatory research, aiming at verifying findings. VM coded all passages in the interview data that focused on (1) data sources used to measure offshore gambling and (2) identified problems and issues in the sources used. These themes also formed the coding framework (data sources; problems and uncertainties). We used a deductive approach because the aim of the key informant interviews was to provide us with additional context to the scoping review results. Coding was conducted using qualitative analysis software Atlas. ti. As the key informant interviews were used as an additional data source in this paper, complementing information from regulator reports in included countries, the results are reported alongside the findings of the results of the scoping review.

**Data from H2 gambling capital**

We used data from H2 to track changes in estimates of the monetary market values of offshore gambling in the Nordics. The aim was to analyse potential uncertainties involved in H2-produced estimates. This was necessary because the methodology for producing these estimates is not publicly available, but the results of the scoping review suggested that H2 is widely used to provide monetary channelling values.

We relied on the H2 database only for this market value analysis, as most other sources included in the review only reported proportionate values or, when monetary estimations were provided, details related to methodologies (such as adjusting to inflation) were missing.

Data from H2 gambling capital are visualised in plots produced with R.

**Research ethics**

According to the guidelines of the Finnish National Board on Research Integrity, no ethics permission was required for this study. The scoping review is based on publicly available information. Inclusion of data directly from H2 gambling capital is based on an existing contract with the Finnish Institute for Health and Welfare. Key informant interview participants were provided information about the aims of the study during recruitment and during the interview. All participants gave informed consent to participate. All participants were also informed that they could withdraw from the study at any time. Participants appear in this study anonymously and no direct quotations are used.

## Results

### Sample characteristics

The full sample of 32 studies and reports, including reference, year of data, context, funder, object of measurement, and data source, is described in Table 2. Offshore measures are produced in each of the four Nordic countries. Most estimates have been published in the Swedish context (N = 13), followed by Finland (N = 8), Norway (N = 7), and Denmark (N = 4). The number of reports has increased over time, with 75% (N = 24) of the studies published in 2019 or after. 17 studies provided timeseries of market developments, while 15 were cross-sections. In line with our inclusion criteria, we only included timeseries data starting from 2010.

In terms of funders for this type of research, the majority has been funded by governmental sources (N = 18), with a high proportion of reports also funded by the industry (N = 11). The highest proportion of industry-funded reports was found in Sweden, where 6 reports were derived from industry sources and 7 from public sources. One Norwegian report was funded by associations that benefit from gambling revenue. Only one Finnish study had received academic funding.

In terms of measurement, 26 reports provided an estimate of the share of GGR that could be attributed to offshore gambling (spending). 12 studies provided an estimate of the population share of individuals engaging in offshore gambling, either in the past year or in the past 3 months. Studies measuring population share were more commonly funded by public sources (11/12 studies), whereas industry funding was more prevalent in studies measuring spending (11/26, all industry-funded reports measured spending).

As for data sources used, 25% of reports (N = 8) used a multimethod analysis, with 75% (N = 24) of reports basing their estimate on a single measure. The most used methods across reports consisted of utilising data from H2 gambling capital (N = 15) or various survey or population study measures (N = 13). One study made use of web traffic data. Other methods consisted of key-informant or stakeholder interviews with industry and other specialists. These were supplementary to other methods. In cases of multiple methods, sales data estimates were typically supplemented with expert interviews, or several sales data estimates were combined. Only two studies [28,34] presented sales figures as well as results from survey responses. In both cases results were presented separately and methods were not integrated.

Across countries, Sweden and Denmark had the highest reliance on H2 figures (6/13 estimates in Sweden, ¾ estimates in Denmark). Finland had the highest reliance on survey methodologies (5/8 estimates). Both industry and publicly funded estimates relied heavily on H2 data. 6/11 industry-funded estimates relied primarily on H2 data, and 8/18 estimates by public instances relied primarily on H2 data. Multimethod analysis was most common in publicly funded reports, especially in Norway.

Our key-informant interviewees also highlighted the importance of multimethod analysis. According to the key-informants, no single data source can capture the full picture of the offshore market and may be biased. Population surveys may be characterised by recollection bias, and company revenue data from offshore operators are estimates rather than based on actual data. It is therefore important to compare and to assess differences. One interviewee also noted that the most reliable data source to measure offshore gambling could be bank transaction data that could track money traffic to offshore gambling websites. Although this type of data can also have limitations, it was not used in any of the studies captured by our review.

### Channelling rate by revenue share

24 of the included studies presented an estimate of channelling based on spending in terms of GGR. The results are presented in Table 3. All studies providing GGR-based estimates reported channelling rates as shares of onshore gambling of the full gambling market. A channelling rate of, for example, 90 percent means that 90 percent of the total GGR is spent on the regulated market.

Included reports varied in terms of what the estimated offshore GGR was compared to. In most cases, estimates were given as shares of the online market or competitive online market. Four studies provided shares of total gambling markets (including land-based gambling) [24,36,43,44]. Five studies provided estimates of both online and total markets [30,33,40,50,53]. Some studies [25,28,42,48–50] also provided estimates for separate product categories. Most separate product category estimates concerned the online sports betting or casino-type gambling markets. When separately mentioned, the channelling rates for these two product groups were systematically lower than the overall channelling rate.

Some reports [25,28] provided several estimates which have been given separately in Table 3. One report [42] used a survey methodology prompting individuals engaging in online gambling on the websites they use. The estimated channelling rate (share of onshore gambling) in the report excludes individuals who did not know whether their website was licensed or not. In Table 3, we have given the estimates both including the 'I don't know (IDK)' responses and excluding them.

19 of the 24 studies expressly mentioned using data from H2 gambling capital in their estimate, either directly or complemented with other data such as stakeholder interviews or 'own analysis'. In addition, two other reports [36,39] used estimates from national authorities that also partly consist of H2 data. Five studies used a population study or survey methodology [26,33,40,42]. One study used a web traffic analysis [25] while one study [28] used a method combining 'seven indicators of total gambling', including helpline contacts, self-exclusions, gambling frequency, historical levels of online gambling, forecasts of online gambling, disposable income, and gross domestic product (GDP).

Overall, channelling rate estimates are lower in monopoly regimes (Finland, Norway, Sweden before 2019). The channelling rate is higher in license-based regimes (Denmark, Sweden after 2019). Different methodologies yielded differing results although these differences could not be systematically assessed due to lack of data. In one Swedish study [28] a population survey methodology yielded higher overall estimates of the share of the onshore market than those derived from H2 data or company revenue. Results obtained using a web traffic methodology also yielded lower estimates of the share of onshore gambling than population survey methods in the estimated years, but not in the years leading up to it [25,42]. In Finland, a population study estimate for 2019 was lower than an estimate based on H2 data for the same year [19,40]. Methodological assumptions are likely to also have an impact [cf. 25]. Similarly, choice of key informants is likely to impact obtained results. Unfortunately, studies utilising key informants did not disclose who these were.

**Table 3. Results of studies measuring channelling rate in terms of revenue share.**

| Source | Context | Methodology | Year | Total market | Online market | Casino | Sports betting | Horse betting | Lotteries | Bingo | Poker |
|---|---|---|---|---|---|---|---|---|---|---|---|
| [24] | Norway | H2 data | 2013 | 92% | | | | | | | |
| | | | 2012 | 93% | | | | | | | |
| | | | 2011 | 93%* | | | | | | | |
| | | | 2010 | 92%* | | | | | | | |
| [25] | Sweden | Webtraffic to unlicensed websites and assumption that spending offshore is **10** times higher than onshore | 2023 (Q3) | | 83% | 74% | 88% | | | | |
| | | | 2022 (Q3) | | 87% | 81%* | 92%* | | | | |
| | | | 2021 (Q3) | | 93%* | 86%* | 95%* | | | | |
| | | | 2020 (Q3) | | 97%* | 95%* | 98%* | | | | |
| | | | 2019 (Q3) | | 98%* | 96%* | 99%* | | | | |
| [25] | Sweden | Webtraffic to unlicensed websites and assumption that spending offshore is **20** times higher than onshore | 2023 (Q3) | | 70% | 59% | 78% | | | | |
| | | | 2022 (Q3) | | 77%* | 67%* | 85%* | | | | |
| | | | 2021 (Q2) | | 85%* | 76%* | 91%* | | | | |
| | | | 2020 (Q3) | | 95%* | 90%* | 97%* | | | | |
| | | | 2019 (Q3) | | 95%* | 92%* | 97%* | | | | |
| [26] | Sweden | Consumer survey (N = 1,014). | 2016 | | 22% | | | | | | |
| [27] | Sweden | H2 data, based on estimate in Copenhagen economics | 2020 | | 75% | | | | | | |
| [28] | Sweden | Method A: H2 and tax data combined with consumer survey on market shares of product categories and average monthly spending per product category. | 2020 | | 81-85% | 72−72% | 80-85% | 98% | 95% | 95% | |
| [28] | Sweden | Method B: Consumer survey (N = 1,000) | 2020 | | 93% | 86% | 99% | 100% | 98% | 79% | |
| [28] | Sweden | Method C: Comparison of revenue development of licensed companies to overall market based on indicators (Sales data, helpline data, exclusion registry, forecasts, disposable income, GDP) | 2020 | | 81% | 72% | 80% | | | | |
| [30] | Denmark | H2 data | 2019 | 91% | 85% | | | | | | |
| | | | 2018 | 90% | 84% | | | | | | |
| | | | 2017 | 88% | 80% | | | | | | |
| | | | 2016 | 87% | 78% | | | | | | |
| | | | 2015 | 86% | 74% | | | | | | |
| | | | 2014 | 84% | 69% | | | | | | |
| | | | 2013 | 83% | 64% | | | | | | |
| | | | 2012 | 83% | 60% | | | | | | |
| [19] | Finland | H2 data | 2019 | 86% | 65% | | | | | | |
| [19] | Finland | Finnish police board estimate | 2019 | | 70% | | | | | | |
| [29] | Norway | H2 data | 2017 | | 45% | | | | | | |
| | | | 2016 | | 45% | | | | | | |
| | | | 2015 | | 45% | | | | | | |
| | | | 2014 | | 46% | | | | | | |
| | | | 2013 | | 43% | | | | | | |
| | | | 2012 | | 50% | | | | | | |
| | | | 2011 | | 45% | | | | | | |

*(Continued)*

**Table 3.** (Continued)

| Source | Context | Methodology | Year | Total market | Online market | Casino | Sports betting | Horse betting | Lotteries | Bingo | Poker |
|---|---|---|---|---|---|---|---|---|---|---|---|
| [31] | Sweden | H2 data | 2019 | | 87% | | 85% | | | | |
| | | | 2018 | | 36% | | 25% | | | | |
| | | | 2017 | | | | 25% | | | | |
| | | | 2016 | | | | 23% | | | | |
| | | | 2015 | | | | 20% | | | | |
| | | | 2014 | | | | 21% | | | | |
| | | | 2013 | | | | 21% | | | | |
| | | | 2012 | | | | 20% | | | | |
| | | | 2011 | | | | 19% | | | | |
| | | | 2010 | | | | 18% | | | | |
| [33] | Norway | Sales data, quarterly survey (N = 1,000), H2 and GBGC data, Swedish tax data | 2022 | 87% | 64% | | | | | | |
| [34] | Norway | Sales data, quarterly survey (N = 1,000), H2 and GBGC data, Swedish tax data | 2023 | | 68% | | | | | | |
| [35] | Norway | H2 data | 2014 | | 45% | | | | | | |
| | | | 2013 | | 53% | | | | | | |
| | | | 2012 | | 53% | | | | | | |
| | | | 2011 | | 50% | | | | | | |
| | | | 2010 | | 44% | | | | | | |
| [36] | Norway | Regulator estimate and sales data | 2021 | 84% | | | | | | | |
| | | | 2020 | 85% | | | | | | | |
| [18] | Sweden | Monitoring of the regulator, H2 data | 2015 | | 46% | | | | | | |
| | | | 2014 | | 43% | | | | | | |
| | | | 2013 | | 45% | | | | | | |
| | | | 2012 | | 48% | | | | | | |
| | | | 2011 | | 47% | | | | | | |
| | | | 2010 | | 47% | | | | | | |
| [39] | Finland | Finnish competition authority estimate | 2022 | | 50% | | | | | | |
| [40] | Finland | Population study of past year gamblers (N = 3,122) | 2019 | 62% | 52% | | | | | | |
| [42] | Sweden | Survey of past 3-month gamblers (N = 3,000), IDK responses excluded from onshore | 2023 | | 77% | 72% | 84% | 89% | 91% | 88% | 72% |
| [42] | Sweden | Survey past 3-month gamblers (N = 3,000), IDK responses included in onshore | 2023 | | 85% | 89% | 95% | 96% | 97% | 94% | 90% |
| [43] | Sweden | H2 data | 2013 | 85% | | | | | | | |
| | | | 2012 | 86% | | | | | | | |
| | | | 2011 | 88% | | | | | | | |
| | | | 2010 | 88% | | | | | | | |
| [44] | Sweden | H2 data with own analysis | 2016 | 77% | | | | | | | |
| | | | 2015 | 79% | | | | | | | |
| | | | 2014 | 80% | | | | | | | |
| | | | 2013 | 83% | | | | | | | |
| | | | 2012 | 84% | | | | | | | |

*(Continued)*

**Table 3.** (Continued)

| Source | Context | Methodology | Year | Total market | Online market | Casino | Sports betting | Horse betting | Lotteries | Bingo | Poker |
|---|---|---|---|---|---|---|---|---|---|---|---|
| [45] | Denmark | H2 data | 2021 | | 90% | | | | | | |
| | | | 2020 | | 88% | | | | | | |
| | | | 2019 | | 88% | | | | | | |
| | | | 2018 | | 87% | | | | | | |
| | | | 2017 | | 84% | | | | | | |
| [46] | Denmark | H2 data | 2023 | | 90% | | | | | | |
| | | | 2022 | | 89% | | | | | | |
| [48] | Sweden | H2 data and key informant interviews | 2020 | | 85% | 75% | 83% | | | | |
| | | | 2019 | | 88% | | | | | | |
| [49] | Sweden | H2 data and Spelinspektionen estimate | 2021 | | 87% | 75% | | 99% | | | |
| | | | 2020 | | 85% | | | | | | |
| | | | 2019 | | 88% | | | | | | |
| [50] | Finland | H2 data and own analysis | 2024e | 65% | 51% | 37% | 29% | | 100% | | |
| | | | 2023e | 67% | 52% | 39% | 30% | | 100% | | |
| | | | 2022e | 69% | 53% | 40% | 32% | | 100% | | |
| | | | 2021 | 71% | 56% | 42% | 35% | | 100% | | |
| [53] | Finland | H2 data | 2023 | 68% | 54% | | | | | | |

* Exact figures not given in the report, estimated based on figure.

Figs 2–5 present country-level comparisons of different estimates across the period of observation (2010–2024). For clarity, industry-funded estimates are shown in black and estimates included in publicly funded reports are shown in red. Although measurements differ and it is impossible to systematically compare results, there are indications that industry-produced estimates of the size of the offshore market are higher than in government-produced estimates. For example, in Denmark, both industry and government estimates are available from 2019–2021. The industry estimates of channelling rates are a few percentage points lower across years (indicating higher offshore market) even though both estimates used H2 data. In Sweden, several government and industry estimates are available from 2020. Excluding the web-traffic analysis, which is experimental, average of channelling rate estimates by government funded reports is 85% and by industry funded reports, the average is 83%.

The key-informant interviews conducted for this study included discussion on how the gambling industry is actively attempting to control the narrative over channelling rates within the Nordic countries, including publishing overestimates of the share of offshore gambling. According to the key-informants, industry actors can use their own estimates and reports on declining channelling rates to, for example, oppose restrictive legislation under the premise that this would lead to increased offshore gambling.

## Channelling rate in monetary terms

Most included studies provided GGR-based estimates as percentages rather than monetary values. When monetary values were provided, these were often directly derived from H2 gambling capital. For this reason, we only provided an estimate of monetary shares of offshore and online gambling using data from H2 gambling capital directly. Fig 6 shows that in monetary terms, measured in GGR, channelling rates can vary as a result of changes within the offshore and the onshore market. For instance, in Finland, the rapid decline in the channelling rate after 2019 (as a percentage of the full market) appears to be a result of declining onshore markets rather than of increased offshore gambling. In Norway, the increased channelling rate after 2021 appears to be a result of declining offshore markets but a stable onshore market.

Channeling rate (GGR) Denmark

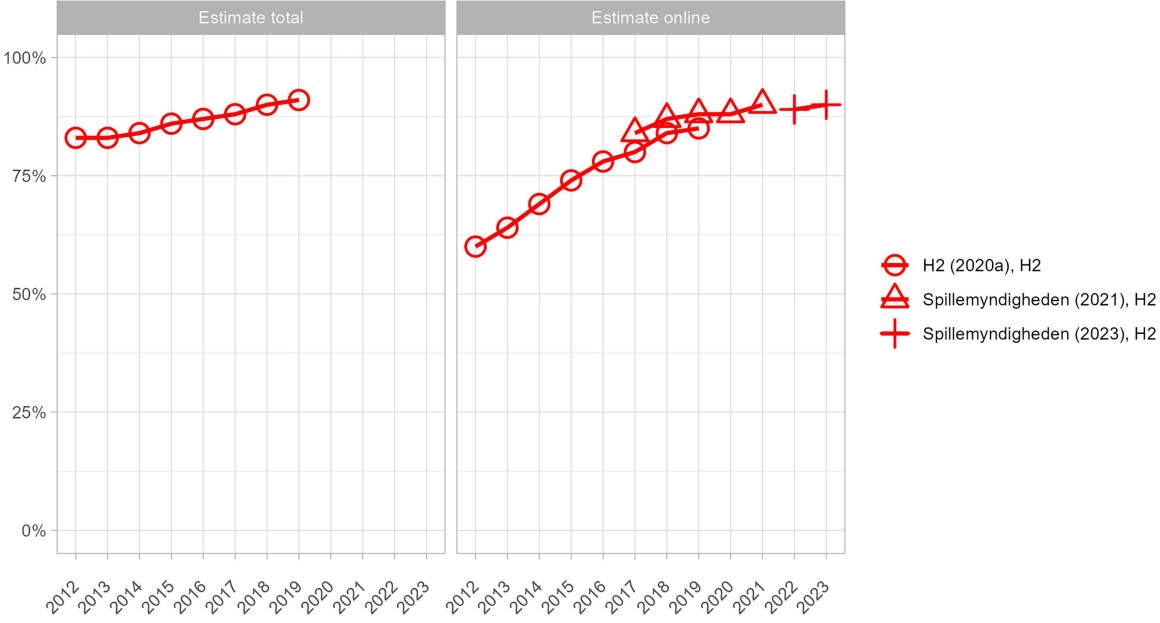

**Fig 2. Different offshore GGR estimates for Denmark.**

Channeling rate (GGR) Finland

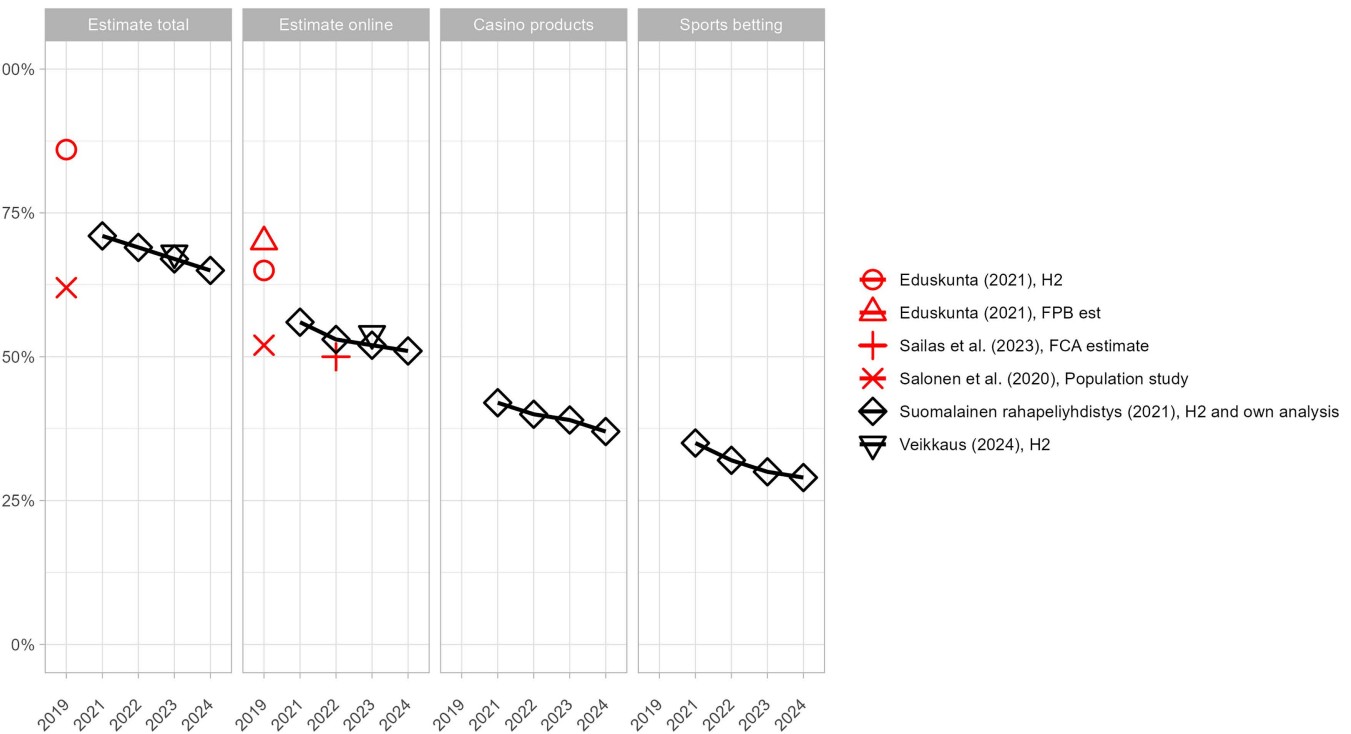

**Fig 3. Different offshore GGR estimates for Finland.**

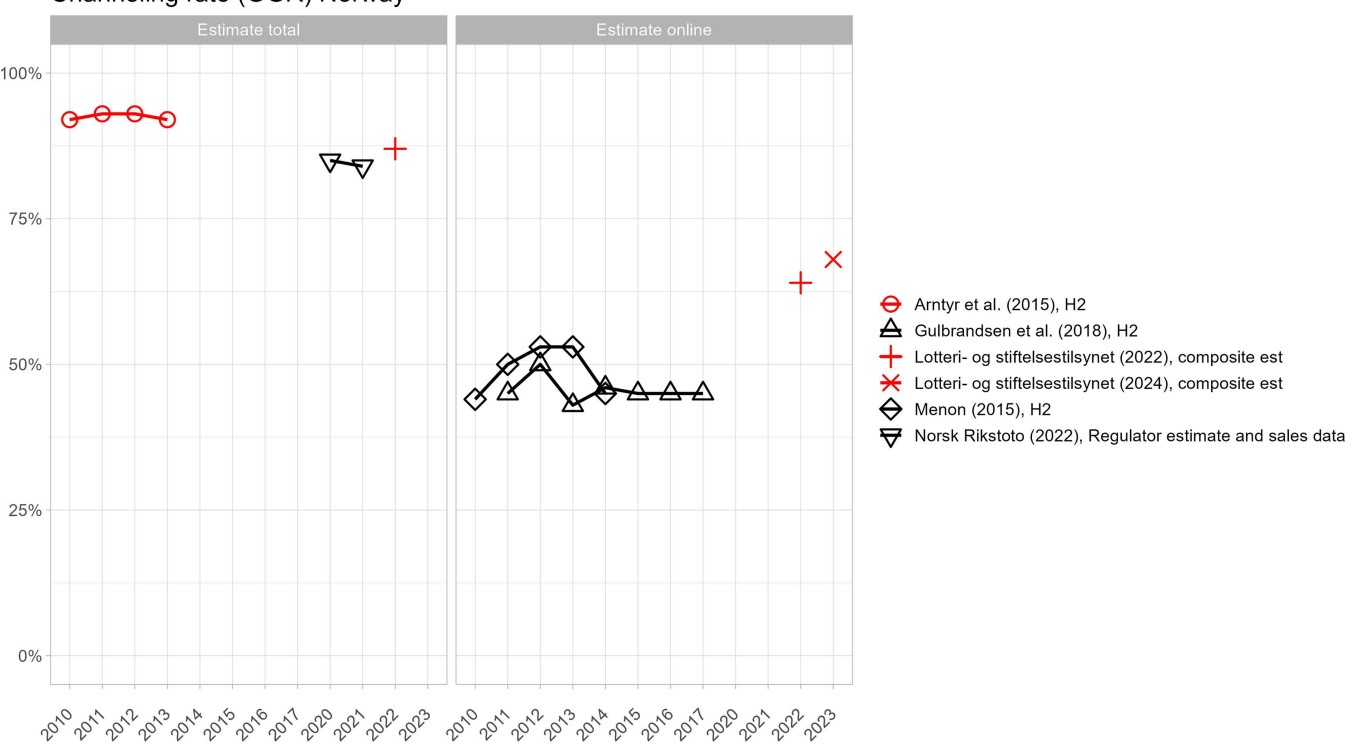

**Fig 4. Different offshore GGR estimates for Norway.**

## Reliance on data from H2 gambling capital

A significant part of studies included in this review made use of data from H2 gambling capital. Whilst this source is likely to be the best available estimate, we also analysed how estimates based on H2 data can vary across time.

The model employed by H2 (as described in [31]) divides gambling market sectors into three categories: onshore (licensed or 'white market'); offshore (gambling licensed in other jurisdictions or 'grey market'); and illegal (unlicensed operation or 'black market'). The channelling rate is only calculated between onshore and offshore markets based on different primary sources including company data, knowledge of the supply side by product vertical, in-house tracking, regular contact with stakeholders and data subscribers, and third-party opinions (including providers and industry analysts). More detailed information on how these estimates are produced could not be found in the reports. In addition, H2 does not provide confidence intervals for its estimates. Our key-informants also highlighted that whilst the H2 database is widely used for estimates, even regulators are not fully aware of metrics and assumptions based on which these estimates are made.

Fig 7 shows our own analysis of how H2 estimates vary over time. For example, estimates for the full offshore market size in the Nordics for the year 2023 (blue line) have fluctuated between over 1,300 million euros in 2022 to less than 1,000 million euros in spring 2023. H2 does not have a regular data update cycle. Instead, figures are updated when the company receives new information that it wants to include in its estimates. Whilst updating estimates based on new information is positive, frequent updates can also include risks. As figures from H2 are used as snapshots in other reporting (such as government reports), these figures remain unchanged in secondary reporting despite updates to the H2 database. Any figure from the H2 database therefore reflects the estimate at that time and may have changed since.

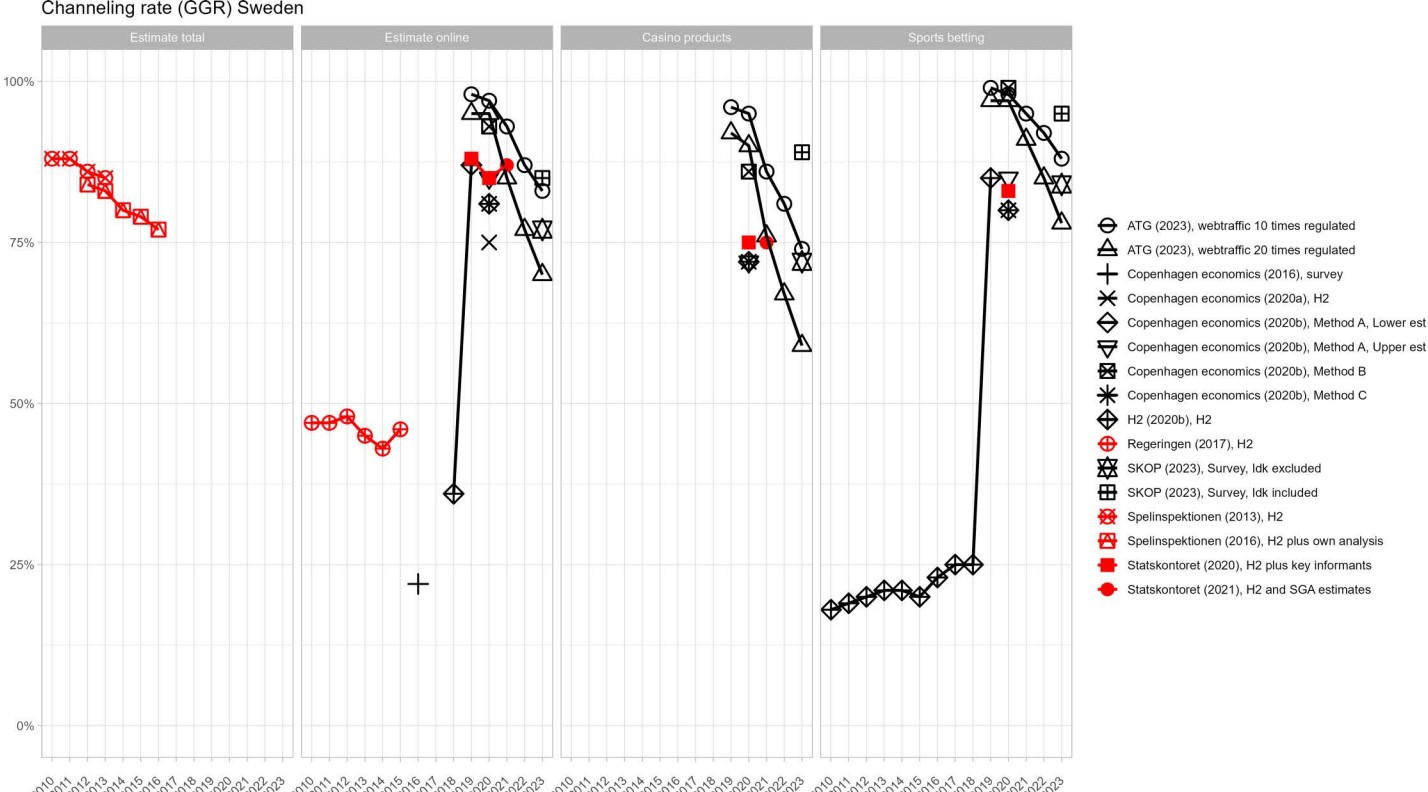

**Fig 5. Different offshore GGR estimates for Sweden.**

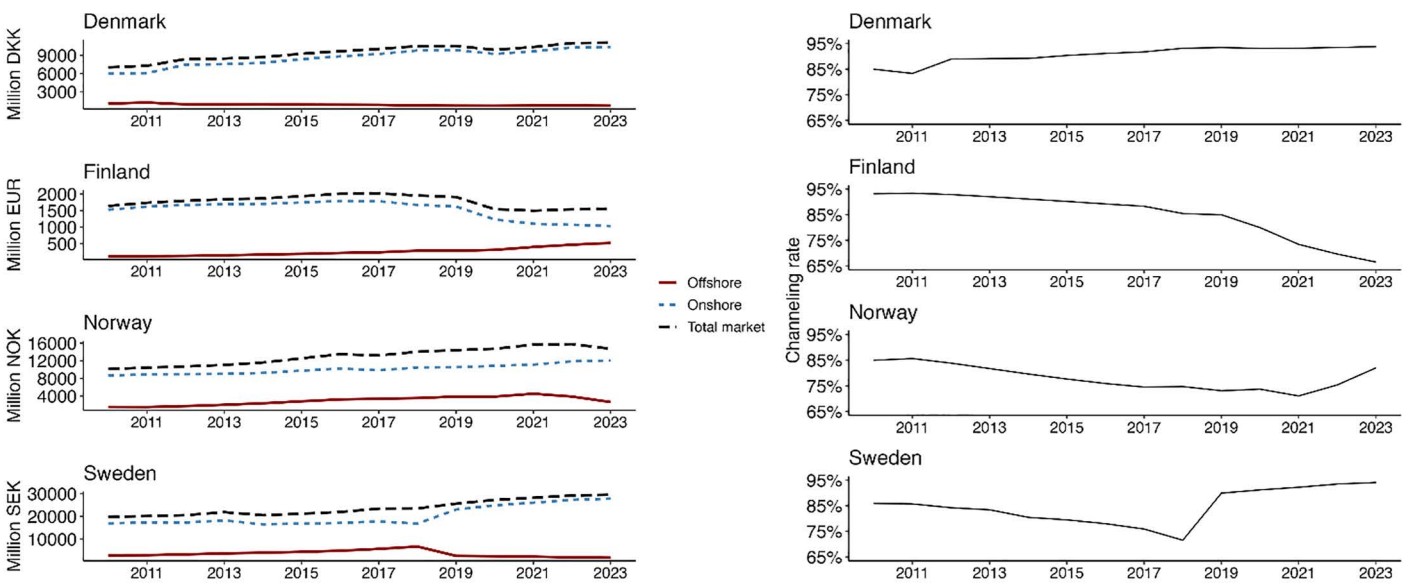

**Fig 6. Channelling rates in Nordic countries in gross gambling revenue (GGR) based on data from H2 gambling capital.**

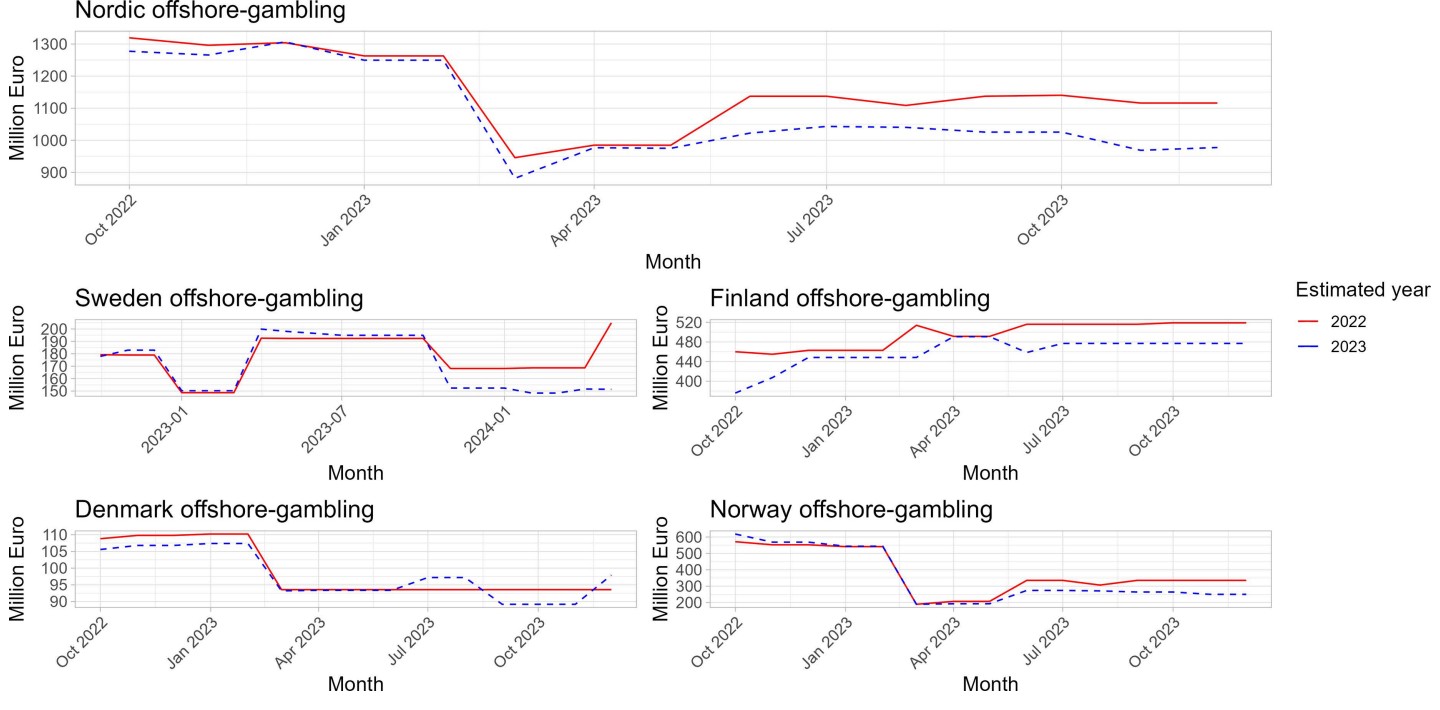

**Fig 7. Variations in offshore estimates provided by H2 gambling capital for 2022 and 2023.**

Frequent updates can also create uncertainty regarding the overall evidence base, as different reports can use different figures for the same time period. Changes do not only concern projections, but historical estimates can also be changed retrospectively. Revised estimates may also be updated again to incorporate different or additional information. However, the database does not provide details on the new data that changes are based on. Furthermore, it is difficult to access estimates that have been changed. This was also the reason why our data covers only 2022–2024. In the absence of a clear reference point for comparison, any estimate can be subject to bias in either direction. Uncertainties in measurement highlight the difficulty in accurately measuring the monetary value of offshore gambling.

### Channelling rate by population share

12 studies provided population share estimates of people engaging in offshore gambling. The studies and their results are presented in Table 4. In contrast to GGR-based estimates that represent the share of onshore gambling in monetary terms, population estimates are given as shares of individuals who participate in offshore gambling. Presenting population and GGR-based estimates in these opposing ways is standard practice in literature. This is likely due to underlying differences in the purpose of these measures. Monetary measures are used for fiscal purposes whereas population-based measures are more common in measuring harm and participation prevalence. As GGR-based and population-based measures are inherently not comparable, we did not convert these figures to align with each other in this analysis.

All population share estimates were produced using a survey methodology. None of the included studies had made use of additional data sources, such as bank transaction data or gambling company data, to verify survey results. Most studies provided an estimate of past 12-month offshore gambling participation either for full population [32,40,51,52] or for the gambling population [37,38]. Two studies provided estimates for those gambling online [41,47]. Overall, and similarly to GGR-based channelling rates, population-based prevalence of offshore gambling was higher in monopoly countries (Norway, Finland) and lower in license-based countries (Sweden, Denmark). Unlike for spending-based measures, we did not

**Table 4. Results of studies measuring channelling rate in terms of population share.**

| Source | Context | Methodology | Year | Full population | Gambling population | Online gambling | Casino | Sports betting | Lot-teries | Horse betting | Bingo | Poker |
|---|---|---|---|---|---|---|---|---|---|---|---|---|
| [33] | Norway | Quarterly survey (N=1,000), past 6 months | 2022 | 3,4% | | | | | | | | |
| [33] | Norway | Quarterly survey (N=1,000), past 3 months | 2022 | 2,5% | | | | | | | | |
| [34] | Norway | Survey (N=2,000), past 6 months | 2023 | 5,0% | | | 12,0% | 5%/12%** | 5,0% | 2,0% | 8,0% | |
| | | | 2022 | | | | 32,0% | 16%/32%** | 9,0% | 1,0% | 29,0% | |
| | | | 2021 | | | | 29,0% | 17%/29%** | 14,0% | 10,0% | 9,0% | |
| [36] | Norway | Figures from regulator (measurements every 6 months) | 2021 | 3,6% | | | | | | | | |
| | | | 2020 | 4,4% | | | | | | | | |
| | | | 2019 | 4,6% | | | | | | | | |
| | | | 2018 | 4,8% | | | | | | | | |
| | | | 2017 | 4,6% | | | | | | | | |
| | | | 2016 | 4,4% | | | | | | | | |
| | | | 2015 | 4,6% | | | | | | | | |
| [36] | Norway | Figures from regulator (measurements every 3 months) | 2021 | 3,6% | | | | | | | | |
| | | | 2020 | 3,9% | | | | | | | | |
| | | | 2019 | 4,1% | | | | | | | | |
| | | | 2018 | 3,9% | | | | | | | | |
| | | | 2017 | 4,6% | | | | | | | | |
| | | | 2016 | 4,7% | | | | | | | | |
| | | | 2015 | 4,0% | | | | | | | | |
| [37] | Sweden | Survey data (N=1,139 gamblers), past 12 months | 2020 | | 3,0% | | 16,0% | 10,0% | | | 9,0% | 17,0% |
| [38] | Norway | Survey (N=1,000 of whom 531 gamblers), past 12 months | 2018 | | 10.0% | | 50,0% | 55,8% | 7,7% | | | |
| [32,40] | Finland | Population study (N=3,077 gamblers), past 12 months | 2019 | 6,2% | | | | | | | | |
| [41] | Sweden | Survey (N=1,002 past 3-month web gamblers), past 3 months | 2021 | | | 7,0% | | | | | | |
| [47] | Denmark | Survey (N=7,636), past 12 months | | | | 3,8% | | | | | | |
| [51] | Finland | Monthly survey (N=2,000 per data collection point), past 12 months | 2022* | | | 5,5% | | | | | | |
| | | | 2023* | | | 5,1% | | | | | | |
| | | | 2024* | | | 5,4% | | | | | | |
| [52] | Finland | Survey (N=1,075), past 12 months | 2022 | 6,0% | | | | | | | | |

* Monthly data collection, annual averages. 2022 and 2024 are incomplete years.

** Data provided separately for sports betting and live betting (latter figures).

find differences between industry and government produced estimates. The only industry-produced population estimate was conducted by the Norwegian gambling monopoly using secondary data.

Three studies also measured participation in different gambling product types [34,37,38]. The studies showed that the participation in offshore gambling was highest in casino-type gambling, sports betting, and poker. Other studies reported results on the most popular offshore gambling products [40,41,47]. In the Finnish context [40], 50.3% of those gambling offshore partook in online EGM gambling, 49% in sports betting, and 27.3% in poker. Online EGMs were the most popular product amongst women, while sports betting was the most popular product amongst men. In Denmark, Spillemyndigheden [47] similarly found that the most popular products amongst those gambling offshore were online casino products (43.3%), sports betting (36%) and skin betting (34.2%). Skin betting involves using virtual items such as video game skins as currency for gambling. SKOP [41] reported that the most common ways of finding offshore offer was by online searches (47%), recommendations (31%), and advertising (31%).

Offshore-type gambling is likely to be easier to measure in surveys produced in monopoly-based regimes as respondents can more easily identify offshore offer. Three studies, each of which were conducted in country with a license-based system [37,41,47] asked respondents whether they can recognise an offshore website. One study [37] conducted in Sweden found that while 3% of the gambling population reported gambling on offshore sites, an additional 17% reported not being sure. Similarly, in the study conducted by SKOP [41], 7% of those gambling online reported offshore gambling, but an additional 12% reported not being sure. In Denmark, the Spillemyndigheden [47] study found that 3.8% of those gambling online reported offshore gambling, with an additional 8.6% not being sure. Furthermore, the study found some other inconsistencies in responses. Notably, some respondents reported no offshore gambling but still mentioned participating in skin betting. There is no licensed skin betting offer in Denmark.

The main reasons given by respondents for participating in offshore gambling included bonuses, better odds, more interesting products, or for some, avoiding a self-exclusion [37,40,41,47]. However, Salonen et al. [40] reported that 98% of individuals reporting offshore gambling also gamble within the regulated market. Amongst these individuals, 37% of total gambling spending took place onshore. This finding suggests that many of those who gamble offshore, also gamble onshore. Offshore gambling is therefore not a separate market segment from onshore gambling.

Oslo Economics [38] asked respondents about their spending on offshore or regulated sites. The results showed that most individuals who gamble in Norway spend moderate sums on gambling, regardless of whether they gamble on the national or on the offshore market. 77% among those gambling offshore reported gambling 200 NOK (17 euros) or less per month. Amongst those gambling with the monopoly, 86% reported gambling 200 NOK (17 euros) or less per month.

## Discussion

This study has reviewed available approaches to and associated uncertainties related to measuring offshore gambling in Denmark, Finland, Norway, and Sweden. Jurisdictions globally differ in terms of how targeted they are by offshore providers, and how prevalent the debate on the size of offshore markets is. The Nordic countries represent a context with high online gambling prevalence that is likely to make them attractive to international gambling providers. Channelling demand to onshore markets is also a key policy argument used in the Nordics. The Nordic region has been subject to many recent policy changes in the gambling field. In the early 2010s, all Nordic countries still operated monopolistic regimes for gambling. Denmark opened its online gambling markets to licensed provision in 2012, Sweden in 2019, and Finland is set to follow in 2027. In each context, the desire to channel consumption from offshore to onshore markets has been a key justification of these policy decisions [18–20].

The Nordic countries therefore represent a context in which measuring channelling and offshore markets is subject to significant political interest. Our results showed that different actors produce estimates of channelling rates across the Nordic region. Estimates are produced using varying methodologies which are often not clearly described. Our investigation of offshore gambling measurement in the Nordics has shown at least three different uncertainties.

First, our results have shown that offshore estimates are likely to be political tools. Policy changes and related debates appear to be connected to the prevalence of published offshore estimates. Within our period of observation (2010–2024), most reports were published in Sweden, followed by Finland. This finding is likely to reflect the licensing regime debates that have taken place, or are taking place, in these countries during the period. Our key-informant interviews also suggested that across the Nordic region, gambling industry stakeholders strive to control the debate over channelling rates by publishing their own evidence. In line with this, an important part of studies captured by our review were funded and published by industry actors (11/32 studies). Most of these were published in Sweden.

The politicisation or even industry capture of offshore estimates echoes prior academic literature on the important role of the gambling industry lobbying for commercial opportunities, under the guise of the offshore or 'black market' threat. A UK study using media reporting found that the industry discussed the black market as an economic threat to the country, with increased regulation of the British gambling industry seen as directly driving consumption to the black market [54]. The industry has also been found to oppose regulation by threatening to leave the regulated market and to establish in an offshore jurisdiction [55]. Leveraging the offshore threat can therefore be seen as part of a larger set of so-called corporate playbook tactics employed by the gambling industry to lobby for favourable regulations [56]. From the perspective of gambling companies, a favourable operating environment can consist of lower taxation rates, more marketing opportunities and reduced social responsibility obligations that can help increase corporate profits.

Second, we found that estimates are largely not comparable because they measure different sides of the offshore market. Measures can focus on the estimated GGR of the offshore industry, or on population prevalence. Population-based estimates can address full populations, gambling populations or even actively gambling populations. GGR-based estimates can be produced as a share of the full gambling field, of the online market specifically, or even for individual online product groups. Depending on what is measured or who are surveyed, produced estimates can vary to a significant degree. For these reasons, the current evidence base does not currently allow for a systematic comparison between methodologies and countries. Although our results suggest that channelling rates appear to be higher in license-based markets than monopoly markets, channelling rates are lowest for products such as sports betting and online EGMs, and that channelling rates can fluctuate, these results need to be interpreted with caution due to existing methodological uncertainties.

For example, surveying individuals who are already gambling online with online casino or sports betting products, yields much higher rates of offshore participation than surveying the full population. Similarly, the inclusion of exclusion of certain product groups in estimates can change results. For example, skin betting or crypto currency betting are not available in the regulated market in any Nordic country, but they are still included in many offshore gambling estimates. Including these products can be justified as novel forms of gambling [57], but they could as well be excluded as they do not compete with regulated offers. From the perspective of measurement, these choices significantly impact obtained estimates.

Offshore consumption typically consists of the most harmful gambling products, including fast-paced online casino products and betting (including live betting). These same products are prevalent in help-seeking statistics in the Nordic region [58]. Offshore websites continue to be available to individuals who have self-excluded from gambling in their national context [59], suggesting that high estimates in these product groups constitute a factor of harm rather than a factor of weak channelling ability. Still, high offshore participation in the online casino and online betting product categories have been the key reason why these harmful product groups have been made highly available in regulated systems to 'compete' with offshore offers.

Third, we found evidence that industry-produced estimates of offshore gambling may be higher than government-produced figures. Industry estimates, particularly in Sweden, also suggested a rapid deterioration of channelling rates. These estimates diverged from governmental estimates. The difference between industry-funded and governmentally produced estimates is likely related to methodological choices as well as interpretation of results. We found that population

surveys and figures based on sales data or web traffic analysis can yield differing estimates. However, due to the size of our sample, we were not able to assess these differences systematically. Instead, we noted some methodological choices that can have an important impact on results and can explain the divergence in estimates. Notably, some industry-sponsored survey studies have chosen to omit 'I don't know' answers from the channelling rate (onshore market), resulting in higher estimates of offshore gambling [42]. Others have made use of novel methodologies such as web traffic analysis, that may be biased due to assumptions that spending on offshore websites would be 10 or even 20 times higher than on regulated websites [25]. Whilst spending on offshore gambling websites may be higher than on regulated websites, we found no empirical basis for these estimates nor conclusive descriptions on how these were determined. Evidence from Norway suggests that for most, sums spent on offshore or onshore gambling are similar [38]. These types of methodological choices and ambiguities were found only in industry-produced evidence.

Overall, our results show that there is no gold standard or one reliable method to conclusively measure offshore gambling. Instead, methodological choices, data resources that have been used, and political interests can have an effect on the kinds of estimates that are produced. Even estimates produced by the same data provider can vary across time.

These results have concrete policy implications. Offshore gambling and available estimates of channelling rates are used as important information sources for policy decisions. Yet, the unreliability of the current body of evidence suggests that more effort needs to be put into improving the scientific quality of offshore gambling measurement. Evidence-based policy should not be based on methodologically ambiguous evidence or estimates that lack transparency.

## Recommendations for a standardised measurement

To improve the evidence base in the field of measuring offshore gambling, it would be necessary to develop a multi-method analysis. The lack of well-developed multi-method approaches was also the most important identified gap in existing literature. This situation has enabled any actor to produce their own estimates based on their own methodological choices. Good, standardised measurement should be based on various data sources and aspects of offshore gambling. Assessment of channelisation cannot be based on a one single measure (GGR- or population share-based) but consider different aspects of channelling. To drive the field forward, we suggest the following steps:

First, the measures included in the indicators need to be calibrated to provide as accurate a picture as possible. For population surveys, this would mean adequately large and representative sample sizes. In addition, it is important to acknowledge that offshore and onshore gambling are not separate market segments as many individuals participate in both [52]. For GGR-based measures, this would include not only considering percentages of full markets, but also monetary estimations. This is because growth in the domestic market can appear as an increasing channelling rate (share of onshore gambling) even if offshore gambling markets would remain constant or even grow but less rapidly in absolute monetary terms. Similarly, reductions in spending to the regulated market can appear as increased offshore gambling in proportion, as has been the case in Finland (also [61]).

Data from H2 gambling capital remains the most used source of information for measuring offshore gambling. However, to improve the reliability of these estimates, the provenance of these figures would require more methodological openness so that they can be scientifically assessed and validated. Web traffic analysis methods also require further developments to reliably measure offshore gambling [also 62]. Web traffic can prove a valuable additional source to measure visitation to offshore sites. However, the use of web traffic data requires making several assumptions regarding gambling behaviours, spending patterns and reasons for website visitation. At least currently, web traffic data is best used as an additional data source rather than as a reliable measure of consumption or even individual-level participation. Scientific research is needed to further develop these data before methodological development can become reliable.

Second, standardised measurement would also need to actively look into new sources of data. In particular, bank transaction data could provide additional insight. In comparison to self-reported spending figures, bank transaction data can allow measuring actual spending to different gambling websites. To also include payments to offshore websites that have been made using

payment intermediaries, data from these third-party services could be used to complement bank transaction data. Bank data can be difficult to access and subject to legal hurdles. Furthermore, collaboration with the full payment processing and banking sector would be necessary [63]. This is not technically impossible as banks are already involved in implementing payment blocking systems [12]. Banks also share data with other applications via open banking. Yet, involving banks in gambling regulation and data provision would likely require a government mandate and require additional legal analysis on legal frameworks and possibilities.

Alongside payment data, more data on the gambling patterns of particular consumer groups are needed. These include, in particular, individuals experiencing harm due to their gambling, such as help-seekers and those who have self-excluded. Statistics from helplines or other help services could be used as an additional data source. Involving lived and living experience could also help capture novel developments in offshore gambling.

Third, standardised measurement would require reliable data that is based on scientific best practice in terms of representativeness, reliability, and transparency. To achieve access to best possible data, formal collaboration across stakeholders, including regulators, payment providers, and web operators is needed. Independent scholars need to have access to different datasets used to produce offshore estimates. By objectively assessing and combining data, it can be possible to arrive at a more comprehensive measures than currently available. Gambling operators and other industry stakeholders need to be mandated to share their data for research purposes. Currently many offshore operators do not disclose any operating data. Access to best available data is also likely to require more collaboration between countries, and particularly across point of sale and point of consumption jurisdictions. This could be possible with increased European Union involvement in gambling regulation and clear government mandates. Optimally, increased collaboration could also capture market shares of 'black market' actors, not only those operating in the grey market as is currently the case for many measures. Leaving 'black market' actors outside of the scope of evaluations will result in inaccurate estimates, particularly as common black market gabling formats such as crypto casinos are becoming increasingly popular [64].

Fourth, it would be important to develop methodologies to move from standardised measurement and indicators to compound measures of GGR and population level offshore engagement. Current state of the art does not permit this type of methodological development. An attempt at a compound methodology was recently published in Sweden [60]. This measure combines estimates based on data from a consumer survey, online traffic (visits and estimated turnover), turnover of gambling software providers, and H2 gambling capital. Methodologically, the report combined these different estimates and produced an average but did not include weights or other specifications. This work could still serve as a basis to developing validated indicators of offshore gambling markets. Based on our review, further developments to this method are still needed. One way to achieve this would be through data linkages across different sources. Data linkages could allow reliably measuring how different estimates perform across individual cases [65].

Fifth and finally, to further improve and consolidate expertise on offshore measurement, more exchange across actors is needed across gambling regulators globally. Our key-informant interviewees reported some informal collaboration in offshore measurement, but no formal structures. Online gambling regulation takes place within the framework of a global online economy. Online gambling operations are expanding rapidly, and in many jurisdictions, regulations are failing to keep up to pace [64]. This globalisation and digitalisation development poses significant risks to effective control of markets and to consumer protection. A global economy is difficult to control with local regulations [also 66] but international bodies, such as the European Union, have no harmonised gambling control measures. Our results have shown that the need for global collaboration is acutely felt in the field of regulating offshore gambling. Going forward, it is therefore crucially important to design a proper implementation framework for all relevant stakeholders to drive forward the development of a more reliable standardised measure.

## Limitations and further studies

Our study has some limitations. Our choice to focus only on the Nordic region for this review yielded a relatively small number of studies (N = 32), some of which reported overlapping data. The heterogenous methodological bases and lack

of openness for estimates do not permit conducting a systematic meta-analysis or even to reliably compare and evaluate existing estimates. A meta-analysis would give a biased and one-sided picture of the channelling rates of Nordic gambling markets. Rather than aiming to synthesise existing estimates, it would be more important to develop more reliable measurement methodologies. In line with the scoping review methodology [22], our paper focuses on scoping methods used in the field and the kinds of uncertainties that are involved in measuring offshore gambling. This means we did not generate a new estimate of offshore market sizes in the Nordics, but rather focused on identifying how existing measures have been produced.

Our analysis was also complicated by the fact that not all underlying studies report their figures in numeric format. This resulted in the need to estimate some of our results from figures. Similarly, many of the included reports did not disclose their methodologies in detail. These limitations are likely to result from the overall lack of academic, peer-reviewed literature in this field, and concomitant high prevalence of grey literature reports with less stringent standards. More methodological openness would help drive the field of offshore measurement forward. Further studies would be needed to study industry impact and ways in which industry actors use the offshore estimates it produces. Further studies should also focus on other stakeholders within this field.

Our focus on the Nordic region means that our results may not be globally representative. Further studies outside of the Nordic context would be needed to test whether similar results are obtained elsewhere. The Nordic model of regulating gambling has traditionally focused on restricting availability via monopoly structures. However, in recent years, the direction of regulation has shifted as the channelling argument has gained more importance [67]. Developments are similar elsewhere. Gambling markets are increasingly global, and offshore gambling can be accessed from any jurisdiction. It is therefore likely that issues relating to measurement of offshore market sizes arise in other jurisdictions as well.

Further studies would also be needed to develop potential composite measures on channelling and to develop the data infrastructures that are needed to support this. Legal studies are needed to assess the data sharing mechanisms and legal frameworks underlying potential data fusion approaches involving, notably, bank data, tax data and individual-level data from gambling companies. More technical research is needed to assess how and whether different data sources can be combined and how these should be weighed. Finally, new approaches are needed to assess the prevalence of black-market gambling. Crypto casino gambling, in particular, can result in inaccurate estimates of the size of the full offshore market. The impact of this type of technical loopholes needs to be assessed and taken into consideration in measurement.

## Conclusions

Based on our analysis of offshore gambling and channelling measurements in the Nordic countries, we infer three conclusions with relevance to gambling policy: First, offshore markets appear as a double-edged sword within regulated markets. Offshore markets can be harmful to consumers, but they can also be used as a tool for regulatory resistance by industry stakeholders. Overestimations of offshore market shares can deter policy makers from effective policy decisions by generating fear of losing control of the market. These choices can further deteriorate public health and exacerbate harm to consumers.

Second, existing measurements of channelling are estimations and vary significantly due to methodological differences, but also due to difficulties in defining what actually constitutes offshore gambling, and which indicators can and should be used to measure it. Policy decisions and debates have therefore drawn from estimates rather than reliable figures. If improved channelling is used as a policy objective, it is crucial to first define how and by whom this is measured. A scientifically validated multi-method measure is needed to improve the evidence-base within the field.

Third, channelling is not only a question of public health, but also of wider public interest. Effectively reducing offshore gambling can protect consumers but also prevent criminal involvement and particularly money laundering. Channelling can also improve data protection. It is therefore important that channelling is accomplished first and foremost by restricting

access to and provision of offshore gambling, rather than by merely increasing spending within the regulated market by making it more attractive. Channelling should not be a policy aim, but rather a means to improved public health, crime prevention, and safer provision by preventing access to harmful offshore gambling offers.

## Supporting information

**S1 File. Prisma scoping review checklist.**
(DOCX)

## Author contributions

**Conceptualization:** Virve Marionneau, Søren Kristiansen, Tomi Roukka, Håkan Wall.

**Data curation:** Virve Marionneau, Søren Kristiansen, Tomi Roukka, Håkan Wall.

**Formal analysis:** Virve Marionneau, Tomi Roukka, Håkan Wall.

**Investigation:** Virve Marionneau, Søren Kristiansen, Håkan Wall.

**Methodology:** Virve Marionneau.

**Project administration:** Virve Marionneau.

**Visualization:** Tomi Roukka, Håkan Wall.

**Writing – original draft:** Virve Marionneau.

**Writing – review & editing:** Virve Marionneau, Søren Kristiansen, Tomi Roukka, Håkan Wall.

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
