## [Decision Letter · Decision Letter 0]

1 Apr 2025

Dear Dr. Marionneau,

Thank you for submitting your manuscript to PLOS ONE. After careful consideration, we feel that it has merit but does not fully meet PLOS ONE’s publication criteria as it currently stands. Therefore, we invite you to submit a revised version of the manuscript that addresses the points raised during the review process.

I recommend that it should be revised taking into account the changes requested by the reviewers. Since the requested changes include valuable and constructive reviews, I would like to give you a chance to revise your manuscript. The revised manuscript will undergo the next round of review by same reviewers.

We look forward to receiving your revised manuscript.

Kind regards,

Baogui Xin, Ph.D.

Academic Editor

PLOS ONE

Journal Requirements:

2. Thank you for stating the following financial disclosure: [VM and TR received funding from the Finnish Ministry of Social Affairs and Health based on provisions of the Finnish Lotteries Act §52. SK is funded by the University of Aalborg. HW received funding from the Swedish Research Council for Health, Working Life, and Welfare (Forte, grant number 2023-00898)].

3. In the online submission form, you indicated that [All data used in the scoping review are fully available from public sources and from the authors. Data from H2 gambling capital are only available under license. These data can be made available by the authors with permission from H2.].

4. Please include captions for your Supporting Information files at the end of your manuscript, and update any in-text citations to match accordingly. Please see our Supporting Information guidelines for more information: http://journals.plos.org/plosone/s/supporting-information .

Reviewers' comments:

Reviewer's Responses to Questions

**Comments to the Author**

1. Is the manuscript technically sound, and do the data support the conclusions?

Reviewer #1: Partly

Reviewer #2: Partly

2. Has the statistical analysis been performed appropriately and rigorously?

Reviewer #1: No

Reviewer #2: Yes

3. Have the authors made all data underlying the findings in their manuscript fully available?

Reviewer #1: No

Reviewer #2: No

4. Is the manuscript presented in an intelligible fashion and written in standard English?

Reviewer #1: No

Reviewer #2: Yes

Reviewer #1: It's an interesting research topic. The authors have discussed the issues of regulations and the gambling market. However, there are some issues that need to be pointed out:

1. The description of offshore gambling is not clear enough. The first thing that must be clearly defined is whether and how offshore gambling is regulated, and what the difficulties are in regulating area.

2. It is suggested that the authors can plot the path of the research method into a block diagram to better understand the research process.

3. In line 404 "Offshore gambling is therefore likely to be highly risky for consumers."—— From an unregulated perspective, the risks of offshore gambling may not be measurable. But from the perspective of social risk management, it is necessary to study the regulation of offshore gambling. It is suggested that the authors can the authors should distinguish between the definition and measurement of risk in the study.

4. The references should be revised.

Reviewer #2: The authors apply a scoping review of offshore gambling and reveal available estimates from academic and grey literature in Denmark, Finland, Norway, and Sweden. They show four key results: (1) policy changes in the related debates may be connected to the prevalence of published offshore estimates; (2) offshore consumption typically consists of the most harmful gambling products; (3) offshore market shares tended to be lower than government-produced figures; (4) no gold standard or one reliable method to measure offshore gambling reliably. It is an interesting approach to estimate hard-to-measurement through literature reviews by tracking databases and shares from the existing literature. There are some issues for consideration:

Related Study Section

Authors should consider adding a short section introducing the use of a scoping review to estimate the scope of the hard-to-measurement industry. Those studies need not be about offshore gambling but scoping reviews. This would help the audience to see the value of adopting a scoping review for this study. For instance, authors should justify using a scoping review rather than a meta-analysis in this section, not the limitation section.

Methodological clarity

The authors analyze the available datasets from their reviews. Table 1: Please provide more content for the national databases and websites in the scoping review for audiences unfamiliar with offshore gambling.

The exclusion of black market shares from the estimates could represent a major hurdle to the final objective of measuring offshore gambling, that is improving public health policy, as stated in the introduction to the study. However, this issue does not appear in the discussion nor in the recommendations sections. As a scoping review’s purpose is to explore an issue and raise new questions for researchers to address, it would be pertinent to encourage analyses covering all forms of online gambling abroad.

As the estimates significantly differ between industry- and government-funded reports, explicitly comparing their methodologies more discriminately could reveal interesting patterns. This could be done not only in the text itself, but also visually in the figures.

On the analysis :

Because the discrepancies between industry and government estimates are significant and have a political motivation, a discriminated analysis would be appropriate, as well as a differentiated visualization.

It should be clarified why the H2 evolving estimates for 2023 are a problem. Adjusting predictions to the latest information is important for increasing estimate accuracy. The real issue is that we don't know what this new information is. That should be made explicit.

Bank data: one of the key informants suggests bank data as the best way to measure GGR and the authors are fully convinced. They include this strategy as a major point in their suggested compound measurement, but they don't discuss this idea in depth.

they say that only one study [27] uses bank data, but in the table study [29] is also listed as using bank data.

they should explain how bank data is used in the reports they mention, how it influences the estimates

they should discuss the drawbacks of banking data, such as issues of accessing the data, which is often protected.

choice of countries: is it not too limiting for a scoping review? The explorative goal of such a study is to gather as many ideas and approaches as possible. Looking beyond these highly similar contexts with tight collaboration would be bring in more approaches to discuss.

The exclusion of black market shares from the estimates could represent a major hurdle to the final objective of measuring offshore gambling, that is improving public health policy, as stated in the introduction to the study. However, this issue does not appear in the discussion nor in the recommendations sections. As a scoping review’s purpose is to explore an issue and raise new questions for researchers to address, it would be pertinent to encourage analyses covering all forms of online gambling abroad.

On the discussions and conclusions

Please state the estimated market size and types of channels of offshore gambling clearly in either the discussions or conclusions. Then, provide limitations of such estimation. Currently, the findings are weak and not focused.

Please use subtitles or highlights to make the findings clearer.

Minor comments:

-          On line 412 in the Discussion section, it is said that the “industry estimates of offshore market shares tended to be lower than government-produced figures”. This is inconsistent with the rest of the paper and is surely just a typing mistake.

-          In the section called “Channeling rate by population share”, a sentence is formulated in a rather confusing manner: “Three studies, both from a country with a license-based system”. I would suggest “Three studies, among which two come from a country with a license-based system”.

-          The last sentence before the Discussion section also lacks clarity. What

**Do you want your identity to be public for this peer review?** For information about this choice, including consent withdrawal, please see our Privacy Policy

Reviewer #1: No

Reviewer #2: No

---

## [Author Response · Author response to Decision Letter 1]

29 Apr 2025

We have uploaded a separate point-by-point response document.

---

## [Decision Letter · Decision Letter 1]

27 Jul 2025

Dear Dr. Marionneau,

Thank you for submitting your manuscript to PLOS ONE. After careful consideration, we feel that it has merit but does not fully meet PLOS ONE’s publication criteria as it currently stands. Therefore, we invite you to submit a revised version of the manuscript that addresses the points raised during the review process.

I recommend that it should be revised taking into account the changes requested by the reviewers. Since the requested changes include valuable and constructive reviews, I would like to give you a chance to revise your manuscript. The revised manuscript will undergo the next round of review by same reviewers.

We look forward to receiving your revised manuscript.

Kind regards,

Baogui Xin, Ph.D.

Academic Editor

PLOS ONE

Journal Requirements:

Reviewers' comments:

Reviewer's Responses to Questions

**Comments to the Author**

Reviewer #3: (No Response)

Reviewer #4: All comments have been addressed

Reviewer #5: All comments have been addressed

Reviewer #6: All comments have been addressed

Reviewer #7: All comments have been addressed

2. Is the manuscript technically sound, and do the data support the conclusions?

Reviewer #3: Yes

Reviewer #4: Yes

Reviewer #5: Yes

Reviewer #6: Yes

Reviewer #7: Yes

3. Has the statistical analysis been performed appropriately and rigorously?

Reviewer #3: Yes

Reviewer #4: Yes

Reviewer #5: Yes

Reviewer #6: No

Reviewer #7: Yes

4. Have the authors made all data underlying the findings in their manuscript fully available?

Reviewer #3: Yes

Reviewer #4: Yes

Reviewer #5: No

Reviewer #6: Yes

Reviewer #7: Yes

5. Is the manuscript presented in an intelligible fashion and written in standard English?

Reviewer #3: No

Reviewer #4: Yes

Reviewer #5: Yes

Reviewer #6: Yes

Reviewer #7: Yes

Reviewer #3: General Comments

This is an interesting and timely study examining offshore gambling using a range of datasets. The topic is clearly important, and the use of multiple data sources strengthens the paper. However, there are several areas where the clarity of writing could be improved, particularly in the introduction and methods sections. A careful read-through is recommended to address grammatical issues throughout.

Introduction

• As this is a multidisciplinary journal, it’s important to clearly define key terms and contextualise the topic early on. Currently, the first half of paragraph one is somewhat confusing, particularly the explanation of offshore gambling. For instance, the statement “Any unlicensed offer can, in practice, constitute offshore gambling” is unclear. Does this mean that a gambling provider can be located within the same country as the gambler but still be considered offshore if unlicensed? If so, consider providing a concrete example to help readers unfamiliar with the regulatory context understand what this looks like in practice.

• Lines 56–58: The sentence here is awkwardly worded. Please revise for greater clarity.

• Line 59: You mention that offshore offers increase competition in the market. It would be helpful to clarify whether this is being presented as a positive or negative effect, or whether it depends on context.

• Line 98: Replace “on” with “across”.

• The final paragraph of the introduction currently includes justification for the scoping review methodology. Consider moving this to the methods section. You could then expand on the broader challenges with existing evidence in the field and clearly state the study’s aims in the final paragraph.

Methods

• You mention that online interviews were conducted, but there is no reference to ethics approval. Was ethics approval obtained, or was a waiver granted? This should be clearly stated.

• The manuscript lacks detail on the data extraction process. Was this conducted by one author or multiple? Was a portion of the extracted data checked by a second author for accuracy? Please clarify.

Results

• Line 248: You note that “key-informant interviewees highlighted the importance of multimethod analysis”, but do not elaborate on what was said to support this. While there is a quote a few sentences later, it’s not clear that this directly relates to the point. Consider clarifying or providing a more direct link between the interview data and this statement.

• Line 373: Please define “skin betting,” as not all readers may be familiar with the term.

Discussion

• Lines 409–414 contain useful background information that would help to frame the paper. I suggest moving this content to the introduction to better set the scene and enhance interpretation of the results.

• Line 459: Replace “unclarities” with “ambiguities”.

• Line 500: The suggestion that banks could provide access to transaction data seems unlikely. Blocking transactions is very different from sharing data. Has this occurred with other sectors or products? It may be worth noting that this would likely require government regulation to be feasible.

• Line 506: Your third point also relies on voluntary collaboration. Historically, private providers rarely share data without a legal mandate. Consider acknowledging this limitation more directly.

Reviewer #4: Premise of research study is focused on public health harm, however, authors do not acknowledge the greater public policy impact of online gambling for the legal-social aspect of data privacy, cybersecurity, and data security impact. May recommend minor revisions to acknowledge this in the project description and future steps/recommendations for frameworks for analysis.

Reviewer #5: This topic is very interesting, but I still have some questions:

1. First of all, the conclusion of this review relies on the screening of the literature, can you clarify the logic and basis of screening?

2, for the main measurements of the comparison of conclusions, can use a table to make a clear show, such as common points or differences.

3, for the interview part, can make some basic introduction, the interview is what the population, whether it is representative, which is crucial to the conclusion of the impact.

4. Can the conclusions of the study be more specific in the abstract, or can some important conclusions be clearly shown?

Reviewer #6: I have carefully reviewed your research and have been deeply inspired by it. Here, I have several questions that I would like to discuss and exchange ideas with you further. I am well aware that some of these questions may not be entirely accurate, but I believe that through our communication, we can gain a deeper understanding of this subject. I look forward to your further response at your convenience. Thank you once again for your contributions to this field!

1.Strengthen Term Definition: The operational distinctions between the “grey market” and the “black market” need to be clearly defined in the introduction, and differences in the Nordic regulatory frameworks should be supplemented (e.g., payment blockades in Finland/Norway vs. website blockades in Denmark).

2.Explanation of Industry Report Motives: The potential motives for industry reports overestimating the offshore market need to be explained (e.g., lobbying for lower tax rates).

3.Argument for Geographical Limitations: The study focuses on the four Nordic countries (Denmark, Finland, Norway, Sweden), but the representativeness for other jurisdictions has not been sufficiently argued.

4.H2 Data Validation: The estimation methods of H2 Gambling Capital data need to be explained (e.g., whether based on transaction flows or company financial reports), and its limitations should be discussed (e.g., not covering the black market associated with crime).

5.Assessment of H2 Data Reliability: 15 out of 32 studies rely on H2 Gambling Capital data, but the transparency of its methodology and the risks associated with industry interests have not been assessed.

6.Exploration of Alternative Data Sources: Objective indicators such as bank transaction data have not been used to verify the survey results.

7.Time Window Selection Justification: The rationale for the time window selection of H2 Gambling Capital data (October 2022 to May 2024) needs to be supplemented.

8.Unified Methodological Standards: The “Methodology” column in Table 3 needs to unify the abbreviation standards (e.g., “H2 estimate” vs. “Webtraffic analysis”); the PRISMA flowchart in Figure 1 should label the specific reasons for exclusion at each screening stage (e.g., the specific criteria for “Not Nordic”).

9.Key Informant Interview Methods: The ethical review of key informant interviews (e.g., signing of informed consent forms); key informant interviews have not specified the sample size, interview guide design, and coding analysis methods.

10.Explanation of Time Span Discrepancy: The literature search covers 2010-2024, but the H2 data analysis only involves 2022-2024, and the impact of the missing earlier data has not been explained.

11.Specific Policy Recommendations: The suggestion in the conclusion to “integrate multiple data sources” needs to be supplemented with specific plans (e.g., integration pathways for bank transaction data).

12.Enhanced Policy Operability: The suggestion to “combine multiple methods and datasets” has not been supported by a specific implementation framework.

13.Bias Risk Assessment: Industry-funded research reports (11 out of 32 items) have not been analyzed for conflicts of interest.

14.Analysis of Stakeholder Impact: The study mentions biases in industry reports but has not analyzed the regulatory mechanisms between regulators and operators.

15.Reflection on Research Limitations: The impact of relying on secondary data on the originality of conclusions has not been reflected upon.

16.Time Window Selection Justification (Duplicate): The rationale for the time window selection of H2 Gambling Capital data (October 2022 to May 2024) needs to be supplemented.

17.Comparative Analysis of Regulatory Effectiveness: The study should compare the regulatory effectiveness between the Nordic countries and other EU countries (e.g., the cross-border impact of Maltese licenses).

Reviewer #7: (No Response)

**Do you want your identity to be public for this peer review?** For information about this choice, including consent withdrawal, please see our Privacy Policy

Reviewer #3: No

Reviewer #4: No

Reviewer #5: No

Reviewer #6: No

Reviewer #7: No

---

## [Author Response · Author response to Decision Letter 2]

22 Aug 2025

Our detailed responses to all comments are in an attached file.

---

## [Decision Letter · Decision Letter 2]

18 Sep 2025

Dear Dr. Marionneau,

Thank you for submitting your manuscript to PLOS ONE. After careful consideration, we feel that it has merit but does not fully meet PLOS ONE’s publication criteria as it currently stands. Therefore, we invite you to submit a revised version of the manuscript that addresses the points raised during the review process.

We recommend that it should be revised taking into account the changes requested by the reviewers. Since the requested changes includes Minor Revision, the revised manuscript will undergo the next round of review by the same reviewers or only by the Academic Editor.

We look forward to receiving your revised manuscript.

Kind regards,

Baogui Xin, Ph.D.

Academic Editor

PLOS ONE

Journal Requirements:

Reviewers' comments:

Reviewer's Responses to Questions

**Comments to the Author**

Reviewer #3: All comments have been addressed

Reviewer #5: All comments have been addressed

Reviewer #6: All comments have been addressed

2. Is the manuscript technically sound, and do the data support the conclusions?

Reviewer #3: Yes

Reviewer #5: Yes

Reviewer #6: Partly

3. Has the statistical analysis been performed appropriately and rigorously?

Reviewer #3: I Don't Know

Reviewer #5: Yes

Reviewer #6: N/A

4. Have the authors made all data underlying the findings in their manuscript fully available?

Reviewer #3: Yes

Reviewer #5: Yes

Reviewer #6: Yes

5. Is the manuscript presented in an intelligible fashion and written in standard English?

Reviewer #3: Yes

Reviewer #5: Yes

Reviewer #6: Yes

Reviewer #3: Well done on responding to the reviewer comments. The paper is greatly improved and will make a great contribution to the field.

Reviewer #5: (No Response)

Reviewer #6: (No Response)

**Do you want your identity to be public for this peer review?** For information about this choice, including consent withdrawal, please see our Privacy Policy

Reviewer #3: No

Reviewer #5: No

Reviewer #6: No

---

## [Author Response · Author response to Decision Letter 3]

2 Oct 2025

Thank you for your excellent suggestions and comments. Responses in separate document.

---

## [Decision Letter · Decision Letter 3]

5 Nov 2025

Dear Dr. Marionneau,

Thank you for submitting your manuscript to PLOS ONE. After careful consideration, we feel that it has merit but does not fully meet PLOS ONE’s publication criteria as it currently stands. Therefore, we invite you to submit a revised version of the manuscript that addresses the points raised during the review process.

We look forward to receiving your revised manuscript.

Kind regards,

Baogui Xin, Ph.D.

Academic Editor

PLOS ONE

Journal Requirements:

Additional Editor Comments:

I recommend that it should be revised taking into account the changes requested by the reviewers. Since the requested changes include valuable and constructive reviews, I would like to give you a chance to revise your manuscript. The revised manuscript will undergo the next round of review by same reviewers.

Reviewer's Responses to Questions

**Comments to the Author**

Reviewer #1: (No Response)

Reviewer #6: All comments have been addressed

Reviewer #8: All comments have been addressed

2. Is the manuscript technically sound, and do the data support the conclusions?

Reviewer #1: Partly

Reviewer #6: Yes

Reviewer #8: Yes

3. Has the statistical analysis been performed appropriately and rigorously?

Reviewer #1: N/A

Reviewer #6: Yes

Reviewer #8: Yes

4. Have the authors made all data underlying the findings in their manuscript fully available?

Reviewer #1: No

Reviewer #6: Yes

Reviewer #8: Yes

5. Is the manuscript presented in an intelligible fashion and written in standard English?

Reviewer #1: Yes

Reviewer #6: Yes

Reviewer #8: Yes

Reviewer #1: The revisions of the manuscript are inadequate.

The research lacks the requisite data validation, and the credibility of the conclusions is insufficient.

Meanwhile, it is recommended that the structure of the manuscript should be re-organized.

Reviewer #6: 1.Add Details on Key Informant Interviews。In the methods section, the criteria for selecting interviewees should be clearly stated (e.g., job level, professional field). The principles for designing the interview outline should also be described, as well as the qualitative analysis method used (e.g., framework analysis or thematic analysis).

2.Add H2 Data Validation Process:。The reliability of the H2 gambling capital data should be verified. This includes methods for cross - validation with other data sources, strategies for seasonal adjustment of time - series data, and specific criteria for classifying gray/black market data.

Reviewer #8: While the topic is timely and the regional focus is well justified, the manuscript reads largely as a descriptive policy review rather than an analytical scientific study. The methodological transparency is limited, the synthesis remains largely narrative, and the discussion offers little conceptual advancement beyond the existing literature. The paper would benefit from clearer methodological framing, more critical comparison across data sources, and a deeper analytical discussion.

**Do you want your identity to be public for this peer review?** For information about this choice, including consent withdrawal, please see our Privacy Policy

Reviewer #1: No

Reviewer #6: No

Reviewer #8: No

---

## [Author Response · Author response to Decision Letter 4]

2 Dec 2025

The response to reviewers document is uploaded as an attachment.

---

## [Decision Letter · Decision Letter 4]

18 Dec 2025

Dear Dr. Marionneau,

Thank you for submitting your manuscript to PLOS ONE. After careful consideration, we feel that it has merit but does not fully meet PLOS ONE’s publication criteria as it currently stands. Therefore, we invite you to submit a revised version of the manuscript that addresses the points raised during the review process.

We recommend that it should be revised taking into account the changes requested by the reviewers. Since the requested changes includes Minor Revision, the revised manuscript will undergo the next round of review by the same reviewers or only by the Academic Editor.

We look forward to receiving your revised manuscript.

Kind regards,

Baogui Xin, Ph.D.

Academic Editor

PLOS One

Journal Requirements:

Reviewers' comments:

Reviewer's Responses to Questions

**Comments to the Author**

Reviewer #6: (No Response)

Reviewer #8: All comments have been addressed

2. Is the manuscript technically sound, and do the data support the conclusions?

Reviewer #6: Yes

Reviewer #8: Yes

3. Has the statistical analysis been performed appropriately and rigorously?

Reviewer #6: Yes

Reviewer #8: Yes

4. Have the authors made all data underlying the findings in their manuscript fully available?

Reviewer #6: Yes

Reviewer #8: Yes

5. Is the manuscript presented in an intelligible fashion and written in standard English?

Reviewer #6: Yes

Reviewer #8: Yes

Reviewer #6: (No Response)

Reviewer #8: This is a solid scoping review with a clear contribution.

However, the interpretation of tables—especially the use of estimated (‘e’) values—would benefit from clearer annotation to improve transparency and reader comprehension.

**Do you want your identity to be public for this peer review?** For information about this choice, including consent withdrawal, please see our Privacy Policy

Reviewer #6: No

Reviewer #8: No

---

## [Author Response · Author response to Decision Letter 5]

20 Dec 2025

We have added a detailed response to reviewers in an attachment.

---

## [Editor Report · Decision Letter 5]

26 Dec 2025

Uncertainties in measuring offshore gambling: A scoping review of Nordic approaches

PONE-D-25-02743R5

Dear Dr. Marionneau,

We’re pleased to inform you that your manuscript has been judged scientifically suitable for publication and will be formally accepted for publication once it meets all outstanding technical requirements.

Kind regards,

Baogui Xin, Ph.D.

Academic Editor

PLOS One
---

## [Editor Report · Acceptance letter]

PONE-D-25-02743R5

PLOS One

Dear Dr. Marionneau,

I'm pleased to inform you that your manuscript has been deemed suitable for publication in PLOS One. Congratulations! Your manuscript is now being handed over to our production team.

Kind regards,

on behalf of

Professor Baogui Xin

Academic Editor

PLOS One